# FutureX: An Advanced Live Benchmark for LLM Agents in Future Prediction

**Zhiyuan Zeng**[1,2,*], **Jiashuo Liu**[2,*,#]

**Siyuan Chen**[2,†], **Tianci He**[2,†], **Yali Liao**[2,†], **Yixiao Tian**[2,†], **Jinpeng Wang**[2,†]
**Zaiyuan Wang**[2,†], **Yang Yang**[2,†], **Lingyue Yin**[2,†], **Mingren Yin**[2,†], **Zhenwei Zhu**[2,†]

**Tianle Cai**[‡], **Zehui Chen**[‡], **Jiecao Chen**[‡], **Yantao Du**[‡], **Xiang Gao**[2,‡]
**Jiacheng Guo**[3,‡], **Liang Hu**[2,‡], **Jianpeng Jiao**[2,‡], **Xiangsheng Li**[‡], **Jingkai Liu**[2,‡]
**Shuang Ni**[‡], **Zhoufutu Wen**[2,‡], **Ge Zhang**[2,‡], **Kaiyuan Zhang**[2,‡], **Xin Zhou**[‡]

**Wenhao Huang**[2,#], **Jose Blanchet**[3], **Xipeng Qiu**[1], **Mengdi Wang**[4]

## Abstract

Future prediction is a complex task for LLM agents, requiring a high level of analytical thinking, information gathering, contextual understanding, and decision-making under uncertainty. Agents must not only gather and interpret vast amounts of dynamic information but also integrate diverse data sources, weigh uncertainties, and adapt predictions based on emerging trends, just as human experts do in fields like politics, economics, and finance. Despite its importance, no large-scale benchmark exists for evaluating agents on future prediction, largely due to challenges in handling real-time updates and retrieving timely, accurate answers. To address this, we introduce **FutureX**, a dynamic and live evaluation benchmark specifically designed for LLM agents performing future prediction tasks. FutureX is the largest and most diverse live benchmark for future prediction, supporting real-time daily updates and eliminating data contamination through an automated pipeline for question gathering and answer collection. We evaluate 25 LLM/agent models, including those with reasoning, search capabilities, and integration of external tools such as the open-source Deep Research Agent and closed-source Deep Research models. This comprehensive evaluation assesses agents' adaptive reasoning and performance in dynamic environments. Our goal is to establish a dynamic, contamination-free evaluation standard that drives the development of LLM agents capable of performing at the level of professional human analysts in complex reasoning and predictive thinking.

## 1 Introduction

The rapid evolution of Large Language Models (LLMs) has catalyzed a fundamental shift in the landscape of artificial intelligence, moving from the generation of coherent text to the creation of autonomous agents capable of complex, goal-oriented behavior (Chase, 2022; Yao et al., 2023; Significant Gravitas; Park et al., 2023; Roucher et al., 2025b). This transition from passive text generators to active problem-solvers necessitates a corresponding evolution in evaluation methodologies. While foundational benchmarks like MMLU (Hendrycks et al., 2021) and SuperGLUE (Wang et al., 2019) are instrumental in assessing the static knowledge of LLMs, they are insufficient for measuring what a model can do when deployed as part of an interactive, goal-seeking system. An agent's performance is defined not just by its underlying model, but by its ability to plan, use external tools, and adapt to a dynamic environment.

---

[*] Project Leaders & Equal Contribution (Junior First).   [#] Corresponding authors.
[†] Core Contributors ($\alpha$-$\beta$ order).   [‡] Contributors ($\alpha$-$\beta$ order).
[1] Fudan University.   [2] ByteDance Seed.   [3] Stanford University.   [4] Princeton University.
Email: {zhiyuanzeng.98, liujiashuo.77, huang.wenhao}@bytedance.com

In response, a new generation of agent-centric benchmarks has emerged, primarily focused on evaluating search (Wei et al., 2025b), tool usage (Yao et al., 2025), and coding skills in controlled or simulated settings (Jimenez et al., 2024). While these benchmarks offer valuable insights into agent capabilities, they largely address static, well-defined problems whose solutions are already known. Further, they fail to address a critical gap: the ability to synthesize dynamic, real-world information, process it, and perform complex analysis and reasoning, which human experts possess across different domains. Future prediction, in fact, directly addresses these two drawbacks. This task directly tests an agent's ability to integrate dynamic, real-world information, process it in context, and generate complex analysis and reasoning about problems whose answers are not yet known to the world. Such tasks naturally involve a dynamic element, and their primary significance lies in preparing agents to anticipate and navigate genuinely novel scenarios, mirroring the foresight applied by human experts across diverse domains.

In response to this need, we introduce **FutureX**, a dynamic and live evaluation benchmark specifically designed for LLM agents performing future prediction tasks. FutureX is built upon a semi-automated pipeline that continuously collects future-oriented questions from 195 diverse websites, curated from a pool of 2,008 sites covering areas such as politics, economics, technology, sports, healthcare, and more. This curation process involves both LLM-based agents and human experts, a necessary combination to ensure quality. Each event is associated with a start date (several days prior to the resolution date) and a resolution date. The pipeline automatically collects and stores agents' predictions on the start date. After the resolution date passes, the system dynamically crawls the web to retrieve the ground-truth outcome and scores the agent's prior predictions. FutureX provides four key advantages that directly address the limitations of existing benchmarks:

- **Large-Scale and Broad Domain Coverage**: Using a semi-automated pipeline for question collection and filtering, we currently select 195 websites from a pool of 2,008 as our sources. These selected websites cover a wide range of topics—including politics, economics, finance, sports, and entertainment—making it, to our knowledge, the *largest and most diverse live benchmark* for future prediction.
- **Real-Time Updates**: FutureX continuously collects future-oriented questions from 128 websites, with daily updates to ensure real-time relevance. By dynamically crawling questions and answers, the benchmark maintains both timeliness and diversity in the questions, presenting a unique challenge for LLM agents to process and adapt to constantly evolving information.
- **No Data Contamination**: Focusing exclusively on future events, FutureX inherently eliminates any risk of data contamination, preventing any exploitation of historical information to manipulate the results.
- **Comprehensive & Automated Assessment of LLM Agents**: Building on FutureX, we have developed a fully automated evaluation pipeline that updates future questions daily, runs various LLM agents for each event on its start date, collects event outcomes after the resolution date, and evaluates agents' performance. The models under evaluation include base LLMs, LLMs with reasoning and search capabilities, open-source Deep Research Agents, and closed-source Deep Research Agents, for a total of 25 models.

In addition to the overall results, we conduct an in-depth analysis, including:

- How do LLM agents perform on questions of varying difficulty levels and across different domains? (Section 4.2 and G.1)
- What factors (such as the type of LLM model, agent framework, and question domain) have the most statistically significant impact on performance? (Appendix G.2)
- How do LLM agents perform when making predictions after the resolution date? (Appendix G.3)
- How planning and search capabilities affect the performance? (Appendix G.4 and G.5)

To pave the way of the "Second Half of AI[1]", we firmly believe FutureX has great potential to unlock new research directions for developing LLM agents capable of performing at the level of professional human analysts in real-world, high-stakes domains.

## 2 RELATED WORK

A significant gap in current LLM agent evaluations is their ability to handle dynamic, real-world information and reason about future events, which are core skills human professional analysts routinely apply across finance, business, politics, and technology trend analysis. Future prediction

---
[1]https://ysymyth.github.io/The-Second-Half/

Table 1: Comparison with Previous Benchmarks for Future Prediction. A ✓ in the Live Update column indicates that a benchmark supports this feature, though may not update regularly. A ✓ in the LLM Agents column for FutureBench reflects evaluation of only a single open-source agent. In contrast, ✓✓✓ to denote regular updates and comprehensive coverage of multiple models.

| | Data | | | | | Evaluation | | | |
|---|---|---|---|---|---|---|---|---|---|
| | #Events | #Domain | Live Update | Time | Source | LLM Agents | Env. | Frequency | Auto |
| ForecastQA (Jin et al., 2021) | 10392 | - | ✗ | Past | 21 News Websites | ✗ | Sim. | One-Time | ✗ |
| Autocast (Zou et al., 2022) | 6707 | 5 | ✗ | Past & Future | 3 Prediction Markets | ✗ | Sim. | One-Time | ✗ |
| OpenEPBench (Guan et al., 2024) | 983 | - | ✗ | Future | 2 News Websites | ✗ | Sim. | One-Time | ✗ |
| NaviTomorrow (Nako & Jatowt, 2025) | 5000 | - | ✗ | Past | 4 News APIs | ✗ | Sim. | One-Time | ✗ |
| ForecastBench (Karger et al., 2025) | 6402 | 8 | ✓ | Future | 4 Prediction Markets 5 Databases | ✗ | Sim. | Monthly | ✓ |
| FutureBench (Together.ai, 2025) | 42 | - | ✓ | Future | 1 Prediction Markets Several News Websites | ✓ | Real | Weekly | ✓ |
| **FutureX** | **∼500/week** | **11** | **✓✓✓** | **Future** | **195 Websites** | **✓✓✓** | **Real** | **Daily Weekly** | ✓ |

serves as a critical test for these capabilities, demanding agents to gather up-to-date evidence and anticipate outcomes in an open-ended environment. However, building a reliable benchmark for forecasting presents unique challenges: unlike static question-answering, future events are inherently uncertain and cannot be easily verified in advance. Previous efforts to evaluate LLMs on forecasting, such as backtesting with historical data (Wildman et al., 2025), risk introducing temporal leakage and retrieval contamination due to retrospective information influencing internet searches on past events (Paleka et al., 2025). Furthermore, many existing benchmarks (Guan et al., 2024; Karger et al., 2025; Nako & Jatowt, 2025) focus on evaluating LLMs without search capabilities, which is impractical for real-world future prediction. Benchmarks like those proposed in Guan et al. (2024) and Nako & Jatowt (2025) are also one-time collected and lack the live updates necessary for dynamic events. While ForecastBench (Karger et al., 2025) attempts to address the "future event" challenge by using only questions about future outcomes, it predominantly evaluates vanilla LLMs, and relies on prediction market events, dominated by multiple-choice questions. This limits both the diversity of events included and, critically, the assessment of an agent's ability to perform open-ended, real-world information gathering. Similarly, FutureBench (Together.ai, 2025) is restricted to events from `PolyMarket` and includes a very small number of events (∼30). This limited diversity in current future prediction benchmarks highlights the inherent difficulty in collecting and evaluating such events, particularly those with unknown answers or those that have not yet transpired.

Although challenging, we adopt this direction because it ensures the absence of information contamination and directly evaluates how effectively an LLM-agent can synthesize real-time data, reason under uncertainty, and predict future events—capabilities that represent the next frontier for expert-level AI agents, which is exactly what our proposed FutureX seeks to achieve. More related works on general LLM agent benchmarks are discussed in Section B.

## 3 FUTUREX

The goal of FutureX is to provide a dynamic, comprehensive, and contamination-free evaluation of LLM agents' advanced search and reasoning capabilities, aiming to match or even surpass the expertise of human professionals. In this section, we will introduce the construction process of FutureX, as well as its core features.

### 3.1 DESIGN PRINCIPLES OF FUTUREX

To clearly articulate the design philosophy of FutureX and draw a sharp contrast with other benchmarks, we demonstrate four core design dimensions.

- **Eliminating Data Contamination.** The design philosophy of FutureX inherently solves the data contamination problem. By defining its core task as "future prediction", FutureX guarantees that the ground-truth answers to all questions have not yet occurred at the time of the agent's prediction, making it impossible for them to exist in any model's training data.
- **Simulating Real-World Challenges and Evaluating Core Intelligence.** FutureX places the agent directly into the real world's information flow (e.g., economic fluctuations, technological breakthroughs, political elections). This task is holistic, demanding a suite of advanced cognitive skills, including information gathering, data synthesis, probability weighing, and causal reasoning.

Figure 1: The overall pipeline of FutureX, which consists of event database construction, future event daily curation, answer daily acquisition. The entire pipeline is fully automated and operates on a daily basis.

- **Large-Scale and Cross-Domain Comprehensive Coverage.** FutureX is currently the *largest and most diverse* live benchmark for future prediction. Through a semi-automated data pipeline, we curate and filter information from 195 high-quality sources, selected from a pool of over 2,000 websites. These sources cover a wide array of domains, including politics, economics, finance, technology, sports, and entertainment.
- **Dynamic and Automated Evaluation Process.** One of the core values of FutureX lies in its highly automated, dynamic, and closed-loop evaluation process. On each event's designated start date, it automatically runs the various agent models and stores their predictions. Once the event's resolution date has passed, the system again automatically crawls the web to obtain the ground-truth outcome and scores the agents' prior predictions. This entire process operates without manual intervention, ensuring the evaluation's timeliness, objectivity, and scalability.

**Overview of FutureX.** Following our design principles, FutureX is a live-updating benchmark for future prediction that covers a broad range of source websites and domains. With daily and weekly updates, it features an automated pipeline for event collection, curation, and agent evaluation—all running smoothly and reliably. As shown in Table 1, FutureX demonstrates clear advantages over previous benchmarks from both data and evaluation perspectives, supporting a much broader range of events and a more diverse set of LLM agents for evaluation. Specifically, recent live benchmarks (Karger et al., 2025; Together.ai, 2025) primarily rely on prediction market websites for live updates. However, as demonstrated in Section 3.3, these events tend to be relatively simple, and many involve subjective questions that are not well-suited for rigorous evaluation. In sharp contrast, FutureX collects and curates events from a much broader range of sources to ensure a challenging and high-quality set of evaluation tasks. Moreover, FutureX evaluates 25 models across four different categories, which, to our knowledge, is the first comprehensive benchmark for LLM agents in the domain of future prediction. In comparison, FutureBench (Together.ai, 2025) evaluates only a single open-source agent with a few LLMs.

## 3.2 Construction of FutureX

As shown in Figure 1, FutureX is an automated, live benchmark that operates on a daily cycle, encompassing four stages: event database construction, future event daily curation, agent daily prediction, and answer daily acquisition. *Each stage is processed on a daily basis.*

### 3.2.1 Event Database Construction

This phase consists of website collection and website curation. During *website collection*, we begin by using the AIME agent (Shi et al., 2025) to gather a large number of website URLs relevant to domains such as politics, economics, finance, technology, and sports, with a total of 2,008 websites. Then for *website curation*, these URLs are then subjected to an initial LLM-based check, utilizing a combination of Seed1.5-Thinking (Seed et al., 2025) and DeepSeek-R1 (Guo et al., 2025). This process performs tasks such as deduplication, assessing the suitability of the websites for question generation, and evaluating their update frequency, reducing the number of websites to 636. The remaining URLs are then manually reviewed, focusing on identifying reliable sources, particularly ranking lists and websites with high update frequency, ensuring that only the most relevant and up-to-date information is included. After this manual review, our initial version has **195** high-quality websites as our current event database, including five types:

- Prediction market websites: Websites that allow users to bet on or predict the outcomes of future events, including political events, sports outcomes, or financial market movements. Examples include `gjopen`, `Polymarket`, etc.

- News websites: Websites that provide up-to-date news, analysis, and market-moving events, such as earnings reports, economic policy changes, sports news, technology trends, and geopolitical developments. Examples include `Reuters`, `ESPN`, etc.
- Entertainment ranking websites: Websites that provide rankings related to music, movies, TV shows, and other entertainment forms. These rankings are often based on user reviews, sales data, critical acclaim, or popularity metrics. Examples include `Billboard`, `IMDb`, etc.
- Government websites: Official websites that provide economic data, regulations, and reports from governments. These include statistical data like GDP, unemployment rates, inflation, fiscal policies, and other public economic indicators. Examples include `U.S. Bureau of Economic Analysis`, `China Agricultural and Rural Information Site`, etc.
- Real-time data platforms: Platforms that provide real-time or near-real-time data on various financial markets. These platforms are used to monitor live stock prices, forex rates, cryptocurrency values, and other asset prices, offering instant updates to help with timely trading decisions. Examples include `Shenzhen Stock Exchange Site`, `Sina Finance`, etc.

In addition, as shown in Figure 1, this event database is updated daily to remove events with unavailable outcomes and continuously add new events using current high-quality websites as "seeds". Based on these, we perform daily future event curation to generate prediction questions for each day, including *event manipulation* and *event filtering*, and details are shown in Appendix C.

### 3.2.2 ANSWER DAILY ACQUISITION

Answer acquisition is a critical phase in FutureX. While we can pose questions about a wide range of future events, the success of evaluation ultimately depends on whether we can reliably obtain the corresponding answers. To this end, much of our effort in constructing the event database (see Section 3.2.1) focuses on ensuring answer availability. We carefully select high-quality websites that consistently provide verifiable outcomes on a daily or weekly basis. Building on this, our pipeline automatically retrieves answers each day by following the procedures outlined below.

1. **Date Filtering.** We filter events whose resolution date aligns with the current day.
2. **Website Crawling.** With the daily answer acquisition set, we then crawl the corresponding website and extract the core content.
3. **Answer Extraction.** Based on the core content retrieved from each website, we use the Seed1.5-Thinking model (Seed et al., 2025) to extract the precise answer.

In addition, as expected during the initial stages, various types of failure cases arise. To systematically address them, we categorize failures into two types: crawling errors and extraction errors. For crawling errors, such as those caused by anti-bot measures, we update our event database to exclude websites that are no longer accessible. For extraction errors, such as incorrect or missing answers, we manually review the cases and design customized prompts to improve answer extraction accuracy. With these efforts, the answer acquisition success rate exceeds **97%**, supporting an efficient and fully automated evaluation pipeline that runs on a daily basis.

### 3.3 DATA OF FUTUREX

**Comprehensive Domain Coverage.** As introduced in Section 3.2, FutureX achieves comprehensive domain coverage through the daily curation of future events from 195 high-quality websites. Between July 20[th] and August 3[rd], our dataset comprises 1,272 events, systematically categorized into 11 main domains—including politics, sports, crypto, culture, finance, business, technology trends, weather, health, and space—each further refined into several sub-categories. As depicted in Figure 10, the distribution across these domains is notably well-balanced, which facilitates a robust and comprehensive evaluation of LLM agents across a multitude of real-world scenarios. We show the examples in major domains in Table 4.

**Different Event Types.** FutureX includes four different event types: single-choice, multi-choice, open-ended ranking, and open-ended numerical prediction events. *Single-choice* events require selecting one correct answer from options fewer than 4. *multi-choice* events involve identifying multiple correct answers, making them inherently more challenging. In addition to events with predefined choices, FutureX features a significant number of *open-ended* events, where no options are provided. There are two types of open-ended events: (1) *ranking* tasks, which typically involve forecasting the order of items on a future leaderboard (e.g., music or movie popularity rankings), and (2) *numerical prediction* tasks, which require estimating a specific numeric value (e.g., a stock price or price index). As detailed in Appendix C, we significantly downsampled binary yes-or-no events—primarily

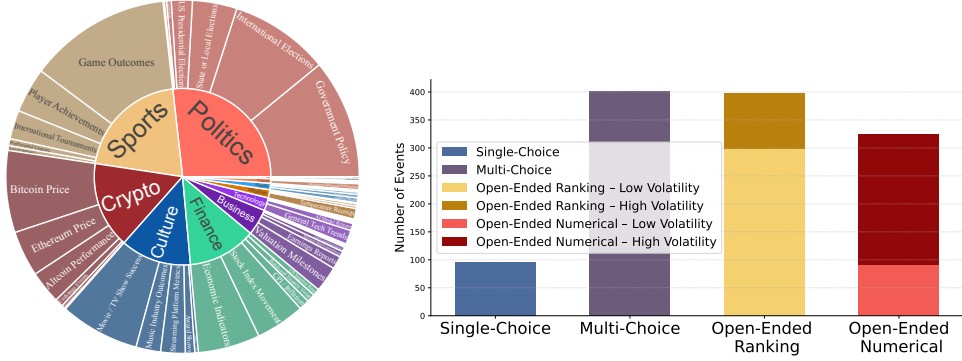

Figure 2: Domains of 1,272 *events*.    Figure 3: Event type distribution.

affecting single-choice questions—to increase the difficulty of the benchmark. As shown in Figure 11, the distribution of the remaining three event types is relatively balanced.

**Different Volatility.**    While all events in FutureX involve forecasting future outcomes, they vary significantly in how dynamic these outcomes are over time. We analyze the *volatility* of each event based on historical data, measuring how much the target outcome is expected to fluctuate over time. We then tag open-ended events as either "Low Volatility" or "High Volatility", which are visually indicated by different color depths in Figure 3.

**Difficulty Tiers.**    Guided by event type and expected volatility (see Figure 11), we partition the benchmark into four progressively harder tiers, basic, wide search, deep search, and super agent, that correspond to the agent capabilities assessed in Levels 1 through 4. An overview of the 4 tiers and the specific agent skills they assess is given in Table 2. The examples of events corresponding to these 4 tiers are shown in Table 3

- The **Basic tier** (Level 1) contains single-choice events with options fewer than 4. The predefined options sharply limit the search space, so information retrieval and reasoning are lightweight.

- The **Wide Search tier** (Level 2) comprises multi-choice events with several correct answers. The agent must submit the full set of valid options and nothing more. This tier requires more complex reasoning, and therefore tests whether the agent can perform *exhaustive yet precise discrimination* across multiple plausible options.

- The **Deep Search tier** (Level 3) contains open-ended events whose underlying facts are relatively stable. With no options provided, the agent must propose its own answer, performing multi-step search and reasoning to gather evidence. Because volatility is low, exhaustive information collection should converge on the correct response. This tier thus probes the agent's ability to *navigate, integrate, and synthesize* reliable information.

- The **Super Agent tier** (Level 4) covers high-volatility, open-ended events. Here the agent must cast a wide net for information and reason probabilistically under shifting signals and deep uncertainty. The task is taxing even for human experts—let alone machines—because the scenarios are complex, ambiguous, and resist simple fact retrieval. This tier therefore probes an agent's "super" capacity for *nuanced, uncertainty-aware forecasting* in the most demanding real-world settings.

Notably, all Level 3 and Level 4 events are generated through our automated pipeline (see Figure 9), which supports scalable event creation while maintaining quality control. This marks a key distinction from prior benchmarks (Karger et al., 2025; Together.ai, 2025), where most events were relatively simple and collected directly from prediction market websites (see Table 1).

## 3.4    EVALUATION PROTOCOL OF FUTUREX

Unlike traditional static benchmarks, where each query is associated with a known answer, future prediction inherently lacks ground truth at the time of prediction, since the relevant events have not yet occurred. As a result, FutureX introduces an evaluation delay, referring to the time gap between when a prediction is made and when it can be evaluated. For example, suppose an agent makes predictions for several events on July 15[th], with resolution dates ranging from July 16[th] to July 22[nd]. In this case, performance can only be evaluated after July 22[nd], once all outcomes are known, resulting in an evaluation delay of one week. FutureX adopts an *one-week* prediction window, which provides both sufficient event coverage and manageable evaluation latency.

Table 2: Difficulty tiers and assessed agent's skills in FutureX.

| Level | Tier | Event Type | Focus | Assessed Agent's Skills | | |
|-------|------|-----------|-------|----------|-----------|-----------|
| | | | | Planning | Reasoning | Searching |
| 1 | Basic | Few choices | Choose from fewer than 4 options from a given list. | Weak | Weak | Weak |
| 2 | Wide Search | Many Choices | Exhaustive discrimination and Return *all* correct futures. | Weak | Medium | Medium |
| 3 | Deep Search | Open-ended (low volatility) | Interactive search & synthesis Navigate sources (click, scroll, filter) Integrate evidence for an answer. | Medium | Medium | Strong |
| 4 | Super Agent | Open-ended (high volatility) | Forecast high-volatility, open-ended events Conduct wide-scope information search Reason and predict under deep uncertainty *"Super-agent"* tier | Strong | Strong | Strong |

Table 3: Examples of different levels, where the specific date can be replaced with any future date.

| Level | Example Events |
|-------|---------------|
| **1. Basic** | Ethereum Up or Down on August 20, 2025? |
| **2. Wide Search** | Who will win the King of the Mountains / Polka-dot Jersey at the 2025 Tour de France A. Tadej Pogacar', B. Other', C. Jonas Vingegaard, ... |
| **3. Deep Search** | Which movies will be in the top 10 of Maoyan Movie Ticketing Rating List as of Beijing Time August 20, 2025. |
| **4. Super Agent** | What is the daily purchase transaction amount (in billion - yuan) in the daily transaction information of the Shanghai - Hong Kong Stock Connect on August 20, 2025, Beijing Time? |

We use the exact-match and F1-score to evaluate the level-1 and level-2 questions, for the level-3 and level-4 questions, we adopt LLM-as-Judge (equation 1) and the clipped MSE score (equation 2) as the evaluation metric. All details can be found in Appendix E.

## 4 EXPERIMENTS

In this section, we present the main results for FutureX from July 20[th] to August 3[rd] [2]. The results in this section cover 25 models, including:

- **Base LLMs (8 models)**: Open-source and closed-source LLMs without tool usage, including *Gemini-2.5-pro*, *DeepSeek V3*, *GPT-4o-mini*, *GPT-4.1*, *DeepSeek R1*, *Qwen3-32B*, *Qwen3-235B*, and *Doubao-Seed1.6-Thinking*.

- **SmolAgent for Deep Research (Roucher et al., 2025a) (6 models)**: SmolAgent is evaluated with various backbone LLMs, including *Gemini-2.5-pro*, *GPT-4.1*, *GPT-4o-mini*, *Qwen3-235B*, *Qwen3-32B*, and *DeepSeek V3*. Among reasoning models, only *Gemini-2.5-pro* is included here, as others—such as *Doubao-Seed1.6-Thinking*, *GPT-o3/o4-mini*, and *DeepSeek R1*—incur significantly longer runtimes and are therefore currently excluded.

- **AgentOrchestra (Zhang et al., 2025b) (2 models)**: Tested with two backbone LLMs—*Gemini-2.5-pro* and *GPT-4.1*. Due to the complexity of this agent framework and limited compatibility, only these representative models are included for now.

- **LLMs (Think&Search) (7 models)**: Evaluation of advanced commercial LLM services with integrated thinking and searching capabilities, including *Doubao*, *DeepSeek R1*, *Hunyuan*, *Qwen3-235B*, *GPT-4o*, *GPT-o4-mini*, and *Grok-4*. *Gemini-2.5-pro (Think&Search)* is on the way.

- **Deep Research Models (2 models)**: Includes top-tier closed-source models tailored for deep research tasks: *Doubao* and *Gemini Deep Research* (with Gemini-2.5-flash).

Note that weekly results (till now) of FutureX online competition are shown in Appendix A, with much more advanced models like GPT5-pro, ChatGPT Agent, Claude 4.1 Opus, Gemini Deep Think, etc.

---

[2]The weekly-updated results after August 3[rd] are shown in Appendix A.

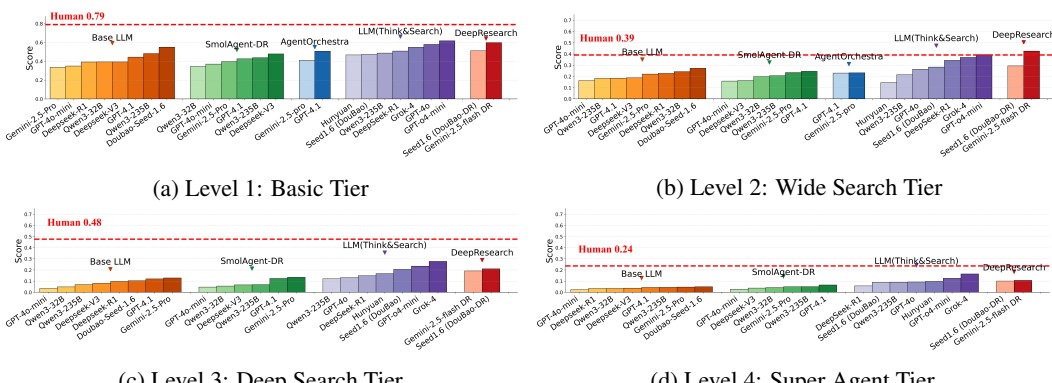

(a) Level 1: Basic Tier  (b) Level 2: Wide Search Tier

(c) Level 3: Deep Search Tier  (d) Level 4: Super Agent Tier

Figure 5: Overall results of different difficulty tiers (between July 20[th] and August 3[rd]). Note that since AgentOrchestra is computationally intensive, we evaluate it with only two backbone models for only Level 1 and 2 events.

## 4.1 OVERALL RESULTS

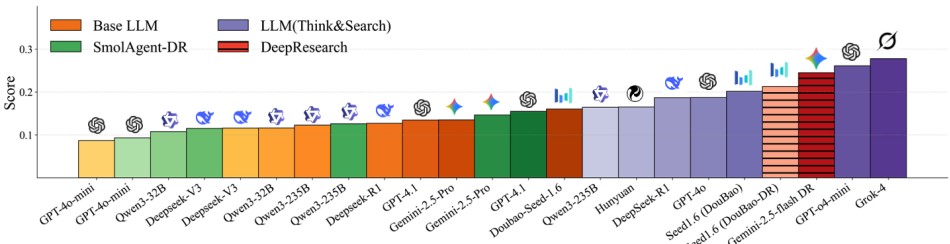

Figure 4: Overall scores on FutureX between July 20[th] and August 3[rd].

As for the overall score, we combine scores from the 4 difficulty tiers (see Table 2) using weights of 10%, 20%, 30%, and 40%, respectively, with heavier weights assigned to the more challenging tiers. The overall results are shown in Figure 4, where models of the same type are represented using similar colors for clarity. As shown in Figure 4, across the four model types, **Grok-4** achieves the highest overall performance, followed by **Gemini-2.5-flash Deep Research**, **GPT-o4-mini (Think&Search)**, and **Seed1.6 (DouBao)**. Generally, reasoning models equipped with search capabilities outperform the rest, underscoring the importance of advanced search and reasoning in FutureX. Moreover, SmolAgent-DR (Roucher et al., 2025a) underperforms compared to LLM (Think&Search), likely reflecting differences in their search API capabilities.

## 4.2 RESULTS ACROSS DIFFICULTY TIERS

We show seperate results across 4 difficulty tiers in Figure 5. Our main findings are as follows:

**Finding 1. Our difficulty tiers accurately reflect the complexity of the events.** We observe a clear, consistent decline in performance across the 4 defined difficulty tiers, which strongly supports the validity of our task stratification. Models achieve high accuracy on Levels 1 and 2, but their performance drops significantly on Level 3 and, in particular, on Level 4. This trend indicates that our difficulty labels effectively correspond to increasing levels of task complexity. Furthermore, by comparing Figure 14 and Figure 15, we find that even within the same domain, model performance declines substantially.

In particular, Level 4 events, which are open-ended and highly volatile, pose significant challenges for current models. These tasks often demand multi-step reasoning, synthesis of ambiguous or incomplete information, long-horizon forecasting, and a deeper understanding of world knowledge and strategic thinking. In our experiments, most models struggle to generate correct responses for these events, and even the strongest models often fail to score at all. In fact, these tasks are so complex that they not only test the limits of current models, but may also serve as a benchmark for measuring superhuman performance in future systems.

**Finding 2. Base LLMs perform well on Level 1 and Level 2 events.** Level 1 and level 2 events are designed as relatively simple single/multi-choice questions. Our results show that even base

LLMs (without tool usage or advanced reasoning capabilities) consistently achieve high accuracy on these tasks. Notably, *DouBao-Seed1.6-Thinking* outperforms several agents equipped with web search tools, including the two Deep Research agents. These events may primarily rely on basic factual recall or straightforward reasoning, which base LLMs are already capable of handling without external tools.

This finding also suggests that Level 1 and Level 2 events are *not sufficiently challenging to distinguish* between models of varying capabilities. While they are useful for establishing a performance baseline, they offer limited insight when evaluating more advanced language models.

**Finding 3. Search/tool usage becomes increasingly important for harder events.** As the complexity of the events increases, particularly in Level 3, models that incorporate external tools such as web search, calculators, or code execution tend to perform significantly better than those that rely solely on static knowledge. This highlights the critical role of tool-augmented reasoning in handling complex, multi-step problems that cannot be solved through pre-trained information alone.

This further demonstrates that FutureX is capable of analyzing advanced search capabilities that are closely linked to reasoning.

**Finding 4. DouBao-Seed1.6-Thinking excels in knowledge retrieval (Level 1 and Level 2), and Grok-4 demonstrates exceptional performance on more difficult events (Level 3 and Level 4).** Among base LLMs, we find that *DouBao-Seed1.6-Thinking* performs the best on Level 1 and Level 2 events. Notably, it even outperforms agents equipped with search tools as well as deep research models. This suggests that, when provided with answer options, *DouBao-Seed1.6-Thinking* is highly effective at retrieving and applying its internal knowledge to make accurate predictions about future events, demonstrating strong inherent reasoning capabilities.

In contrast, among all evaluated models, *Grok-4* stands out on the most challenging tasks. Remarkably, it surpasses even premium models such as Gemini Deep Research in both accuracy and efficiency. Despite operating with fewer searching and faster inference speeds, *Grok-4* and *GPT-o4-mini* achieve top-tier results, highlighting an impressive balance between reasoning strength and runtime efficiency.

**Finding 5: LLM agents still lag behind humans.** In addition to automated model evaluations, we conducted a human annotation study with 40 industry experts. These participants comprised current or former employees from the Big Four accounting firms (e.g., KPMG), top consulting firms (e.g., McKinsey), and nine leading investment banks (e.g., UBS). We randomly sampled 300 questions from our test bank and had these experts answer them independently, then computed their average scores on the same evaluation metrics (red dotted lines).

As shown in Figure 5, humans significantly outperform LLM agents on Level 1, Level 3, and Level 4 events, indicating that LLMs still have considerable ground to cover before matching human expertise. Conversely, for Level 2 events, some models actually surpass human performance; this may be because these multi-choice questions involve so many options that people often cannot exhaustively compare every possibility. Overall, these results underscore the substantial potential for LLM agents to aid (and eventually rival) humans in forecasting future events.

## 4.3 MORE CASE STUDIES

In addition to our main results, we design and conduct multiple case studies, including:

1. **Results across Different Domains.** Differnt models have different strengths, for example, GPT models are good at Crypto. Details in Appendix G.1.

2. **Factor Analysis.** We conduct a linear analysis to discover important factors with statistical significance. From it, the "difficulty level" and "event domain" factors are identified as the most critical factor that influence the performance of agents. Details in Appendix G.2

3. **Past vs. Future Prediction.** By comparing historical event searching and future prediction, we could directly characterize the search capability of different models, where Grok-4 exhibits excellent performance. Details in Appendix G.3

4. **Analysis of Agent Trajectories.** The number of tool-call, comprehensiveness of planning and the credibility of source websites all significantly influence the agents' performance in future prediction. Details in Appendix G.4

5. **Search Analysis.** We count the average number of searches for each model, where we find strong models typically conduct more searches, and Grok-4 searches even more than deep research models. Details in Appendix G.5

More recent results of our weekly competition are shown in Appendix A, and a comprehensive review of related works can be found in Appendix B.

# 5 CONCLUSION

FutureX is the first live benchmark that tests LLM agents on real-world future prediction tasks. Going forward, FutureX offers a flexible platform for improving LLM agents. We are actively working on adding new domains and data sources to FutureX. By keeping the benchmark live and diverse, we aim to push agents closer to the level of human experts in making timely, strategic predictions across a wide range of fields.

# 6 ETHICS STATEMENT

This work did not involve animal subjects or personally identifiable data. All information used are publicly available and were accessed in accordance with their licenses and terms of use.

# 7 REPRODUCIBILITY STATEMENT

All information necessary to reproduce our results is included, covering benchmark construction procedures, the test date, evaluation metrics, and the set of models under evaluation.

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

## A  WEEKLY RESULTS OF THE FUTUREX ONLINE COMPETITION

Following the release of FutureX, we launched a weekly online competition that has attracted participants from both industry and academia, including MiroMind[3], Foresight[4], Peking University, as well as many independent researchers and practitioners.

In the main body, we report results on 1,272 events conducted between July 20[th] and August 3[rd]. In this appendix, we provide complementary results on a weekly basis using more recent data. In order to lower the participation bar, we sample ∼100 events every week for the weekly test. For the weekly competition results, in addition to the models evaluated in the main body, we also include GPT-5 Pro (web), ChatGPT Agent (web), Gemini Deep Think (web), and Claude 4.1 Opus.

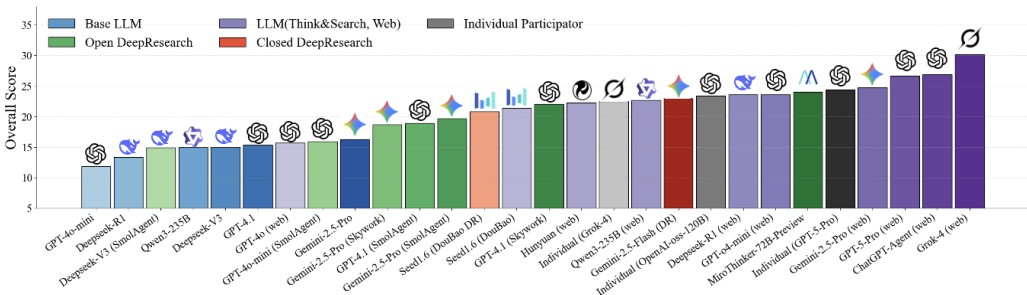

Figure 6: Results of events between August 27[th] and September 2[nd].

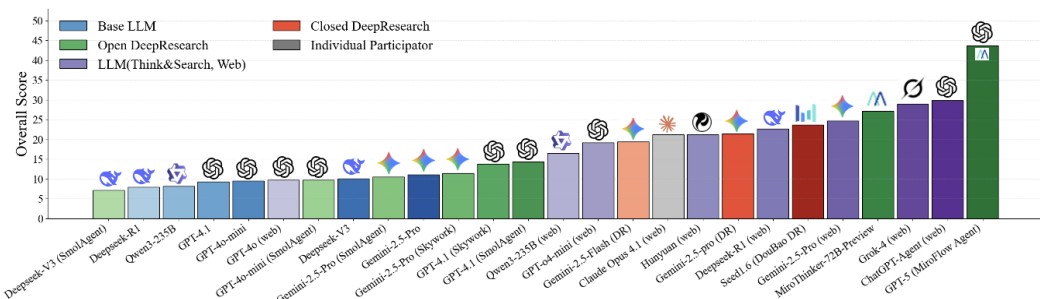

Figure 7: Results of events between September 3[rd] and September 9[th].

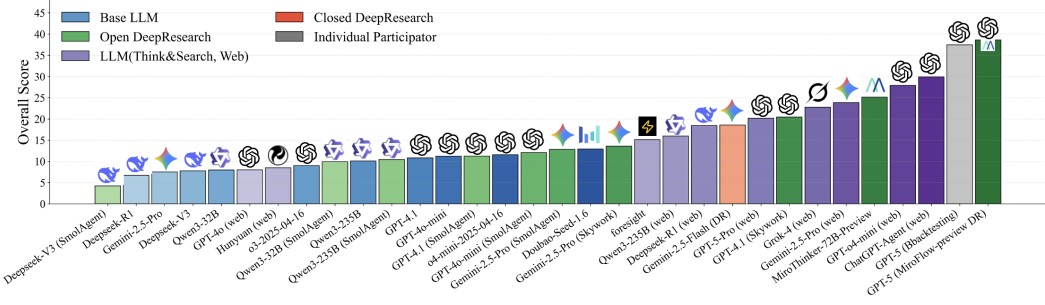

Figure 8: Results of events between September 10[th] and September 16[th].

[3] https://miromind.ai/
[4] https://foresight.lightningrod.ai/

As the competition progresses, we observe that purpose-built agents, such as MiroFlow with GPT-5 and the Bbacktesting Agent with GPT-5[5], consistently outperform general LLM web products and deep research agents. This finding indicates that the task of future prediction is, to some extent, both *learnable* and *optimizable*. Moreover, such results suggest the potential for agents to eventually surpass human experts, further underscoring the significance of our FutureX benchmark.

# B   COMPREHENSIVE RELATED WORK

In addition to recent benchmarks for future prediction (see Section 2), this section introduce relevant literature on LLM agent benchmarks, encompassing both established and recent live evaluations.

**Agent Benchmark.**   A new wave of benchmarks is designed specifically to evaluate LLM-based agents on complex, interactive tasks rather than isolated questions. For example, AgentBench (Liu et al., 2024) introduces 8 distinct simulated environments (from operating systems and databases to web interfaces and games) to assess an agent's planning, tool use, and decision-making abilities. WebArena (Zhou et al., 2024) provides high-fidelity simulations of real websites (e-commerce, forums, collaborative coding, content management) to test long-horizon web interaction tasks, where Agents must execute multi-step web browsing actions to accomplish user goals. Another benchmark, GAIA (Mialon et al., 2023), focuses on general-purpose assistant capabilities with 466 real-world questions that require reasoning, multi-modality, web search, and tool use. In the realm of information retrieval, BrowseComp (Wei et al., 2025a) consists of 1,266 "inverted" questions designed to entangle information and thwart simple lookup. Agents must perform creative, multi-hop web searches to uncover hidden facts, testing their persistence and strategic search skills beyond basic fact retrieval. In software engineering, SWE-bench (Jimenez et al., 2024) evaluates agents on resolving real GitHub issues by generating code patches and verifying them against project test suites, and based on this, multiple variants are proposed, such as SWE-bench-Verified (OpenAI, 2024), SWE-gym (Pan et al., 2025), Multi-SWE-bench (Zan et al., 2025), and SWE-smith (Yang et al., 2025). Collectively, these benchmarks offer valuable insights into various aspects of agent performance—from web navigation and tool use to coding—but they primarily operate in controlled environments with predefined task scopes and information. Additionally, these benchmarks do not integrate complex reasoning and information-gathering capabilities, both of which are essential for our proposed FutureX.

**Live Benchmark.**   Besides traditional benchmarks with static question sets, live benchmarks have recently emerged, such as LiveBench (White et al., 2025), LiveCodeBench (Jain et al., 2024) and SWE-bench-live (Zhang et al., 2025a), which automatically update questions to enable more reliable, contamination-free evaluations. Besides, Li et al. (2024) introduce Arena-Hard that is frequently updated from live data in Chatbot Arena (Chiang et al., 2024) to avoid potential over-fitting or test set leakage. Our proposed FutureX follows this trend, offering a fully automated, live benchmark for future prediction—where even the events themselves are live. Compared to the benchmarks discussed, ours is more aligned with real-world professional analysis scenarios across multiple domains, making it highly practical.

---

[5]These two agents are given by the participators and currently are not open-sourced.

## C    DETAILS ABOUT DATA CURATION AND FILTERING

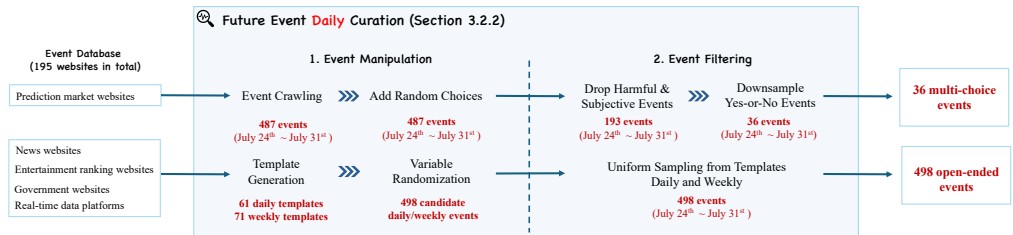

Figure 9: The detailed future event daily curation process, which manipulates daily and weekly events from the event database. As an example, the number of events between July 24th and July 31st remaining after each step is shown in red. Note that the number of events each week varies due to fluctuations in prediction market events (the first row in the figure).

Based on the event templates, we perform daily future event curation to generate prediction questions for each day. As shown in Figure 9, this process consists of two main steps: event manipulation and event filtering.

**Event Manipulation**    This phase involves transforming each website into a concrete future event format suitable for our pipeline, which varies depending on the type of website.

**Prediction market websites**    There are already feature future prediction events on prediction market websites. Therefore, we crawl these events on a daily basis, which are typically *binary or multiple-choice*. For each event, we employ the Seed1.5-Thinking model (Seed et al., 2025) to introduce a set of unrelated (random) choices, thereby increasing the complexity and challenging the system's search and reasoning capabilities at a higher level. For instance, if LLM agents were to query each choice individually, it would significantly reduce efficiency.

**Other types of websites**    For other websites where future events cannot be directly obtained, we follow these steps to make more challenging *open-ended questions*:

- Template Generation: We create an "event template" for each site, which can take variables (for example, target, date, etc.) as inputs to make the events adaptable over time, even for the same website. The process is as follows: First, we use an LLM to design candidate question templates based on the content of each webpage. Next, we specify the input variables for answer scraping. After the answer is scraped, the LLM checks whether the answer can be successfully retrieved. To ensure reliability, a human reviewer then verifies the results and selects the most appropriate question templates. Once an event template is established, it remains fixed within our pipeline, eliminating the need for recreation in subsequent iterations. Below are two examples.

  > Website 1: `https://www.dongchedi.com/` (China's largest car review website)
  > – Variables: `rank`, `date`, `target`
  > – Event Template: Which car will be ranked {`rank`} on the {`target`} board on {`date`} at Dongchedi?
  >
  > Website 2: `https://www.google.com/finance/`
  > – Variables: `stock`, `date`
  > – Event Template: What will be the highest point of {`stock`} on {`date`}?

- Randomization: Based on the event templates, in order to prevent asking the same future event every day, we apply randomization to our templates by varying the input variables within the same event template (and for the same website). For example, for ranking websites, we may ask the LLM agents to predict different ranks each day or predict ranks within different sub-ranking lists. For government websites, we may request different statistics or

metrics. For real-time data platforms, we may focus on different markets, indexes, or stocks. Below are some examples.

---

Template 1: Which car will be ranked {`rank`} on the {`target`} board on {`date`} at Dongchedi?
- Q1. Which car will be ranked *1st* on the *SUV Popularity Ranking* board on *September 1st* at Dongchedi?
- Q2. Which car will be ranked *3rd* on the *MPV Sales Ranking* board on *September 15th* at Dongchedi?

Template 2: What will be the highest point of {`stock`} on {`date`}?
- Q1. What will be the highest point of *APPLE* on *September 1st*?
- Q2. What will be the highest point of *NVIDIA* on *September 7th*?

---

After manipulating the events, we are able to generate ∼**500** daily and weekly future events as candidates out of 195 high-quality websites. Note that each future event is associated with an answer resolution date that will be used in the Answer Daily Acquisition phase (see Section 3.2.2).

**Event Filtering**   For events crawled from the internet (for example, prediction market websites like `Polymarket` and `gjopen`), we carefully filter the event set before testing the LLM agents to ensure the validity of the events, where we mainly filter out easy or trivial events, harmful events, and subjective events.

- *Harmful events*: These events include content that may involve discrimination, hate speech, or other harmful factors. Such events can introduce bias or propagate misleading information, undermining the integrity of the predictions. To mitigate this, we use a combination of Seed1.5-Thinking Seed et al. (2025), DeepSeek-R1 Guo et al. (2025), and Gemini-2.5-flash (Comanici et al., 2025) to filter out harmful events from the set before testing to ensure that only appropriate and reliable data is used.

- *Subjective events*: Events that rely on individual opinions or subjective judgment are difficult to predict reliably. These events introduce significant variability in responses, which can disrupt the testing process. To address this, we use LLM-as-a-judge to filter out such events with a combination of Seed1.5-Thinking Seed et al. (2025), DeepSeek-R1 Guo et al. (2025), and Gemini-2.5-flash (Comanici et al., 2025) to ensure reliability. As shown in Figure 9, between July 15th and July 22nd, we drop 294 unsuitable events (both harmful and subjective events). Examples include: "Will we win 100k tomorrow at the mara hackathon?" and "I finish Park's "Our Nation's Path" by EOM July?".

- *Yes-or-No events*: Events with binary choices (such as yes or no, or the outcome of a single match) are relatively easy to predict, with even random guessing achieving an accuracy of 50%. Since we cannot introduce additional choices for these events, we significantly downsample these yes/no binary events. As shown in Figure 9, events collected from prediction market websites are reduced from 193 to 36 for one week.

Through event filtering, we significantly reduce the number of events from prediction market websites to make our benchmark more challenging. This ensures the high quality of FutureX and stands in sharp contrast to previous benchmarks (Together.ai, 2025; Karger et al., 2025), where prediction market events dominate.

Additionally, to maintain event diversity and prevent homogeneity across other types of websites, we randomly select only one question per template per website for inclusion in the daily prediction set, resulting in 61 daily events and 71 weekly events (61x7+71=498 open-ended events every week).

## D   DETAILS ABOUT FUTUREX DATA

**Comprehensive Domain Coverage**   As introduced in Section 3.2, FutureX achieves comprehensive domain coverage through the daily curation of future events from 195 high-quality websites. Between July 20th and August 3rd, our dataset comprises 1,272 events, systematically categorized into 11 main domains—including politics, sports, crypto, culture, finance, business, technology

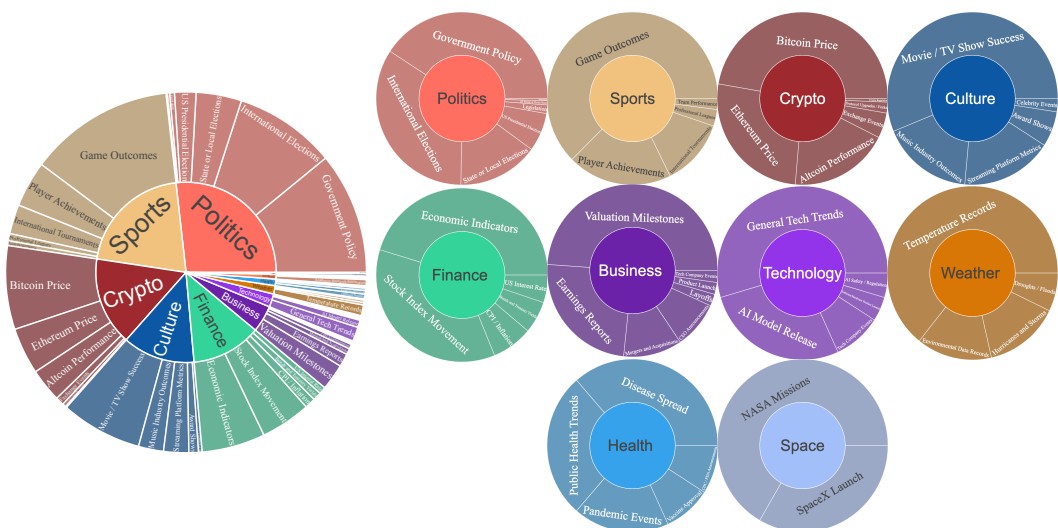

Figure 10: Detailed domains of all *events* included in FutureX, from July 20[th] to August 3[rd], total 1,272 events. These events are organized into 11 main categories—politics, sports, crypto, culture, finance, business, technology, weather, health, and space—with each category containing several sub-categories.

trends, weather, health, and space—each further refined into several sub-categories. As depicted in Figure 10, the distribution across these domains is notably well-balanced, which facilitates a robust and comprehensive evaluation of LLM agents across a multitude of real-world scenarios. This comprehensive domain coverage provides two key advantages:

1. It enables a holistic assessment of LLM agents' overall future prediction capabilities, as diverse fields often necessitate distinct analytical approaches and specialized reasoning strategies.

2. Coupled with FutureX's live updating, this breadth makes our benchmark significantly more robust against exploitation or overfitting, and we anticipate this will ensure it remains a challenging and relevant evaluation for the foreseeable future.

We show the examples in major domains in Table 4.

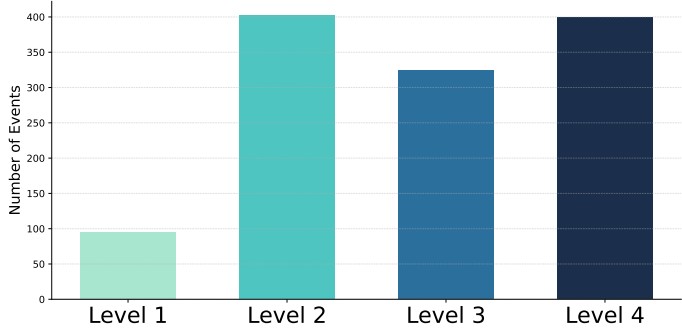

Figure 11: Difficulty level distribution (between July 20[th] and August 3[rd]).

Table 4: Examples to be Predicted by Domain. We take the date August 20, 2025 as an example, which can be replaced with any time in the future.

| Domain | Event to be Predicted |
|---|---|
| **Culture & Media** | Please predict the Top 10 Gross in US dollars on Box Office Mojo's Daily Box Office for August 20, 2025, Beijing Time. |
| **Finance & Economy** | Please predict the grain price index within the 'Agricultural Product Wholesale Price 200 Index' from the National Agricultural Product Wholesale Market Price Information System for August 20, 2025, Beijing Time. |
| **Technology** | Please predict the market share percentage of Win10 in the operating system rankings published by 51.LA for August 20, 2025, Beijing Time. |
| **Crypto** | Please predict what the Fear and Greed Index on CoinMarketCap will be on August 20, 2025, Beijing Time. |
| **Business & Companies** | Please predict which models will be in the top five of Dongchedi's national popular sedan rankings for August 20, 2025, Beijing Time. |

## E DETAILS ABOUT EVALUATION METRIC

As introduced in Section 3.3, we have multiple types of events in the benchmark, each with a different evaluation metric. As for single-choice events, the metric is simply the 0-1 error:

$$\text{score}(Y, \hat{Y}) = \mathbb{I}(Y = \hat{Y}).$$

For multi-choice events, as the answer contains multiple right options (denoted by $\mathcal{Y}$), the metric is as follows:

$$\text{score}(\mathcal{Y}, \hat{\mathcal{Y}}) = \text{F1-Score}(\mathcal{Y}, \hat{\mathcal{Y}}).$$

For open-ended ranking events, such as predicting the top-$k$ ranked items, we treat the ground truth as an ordered list, denoted by $\{y_1, \ldots, y_k\}$. To evaluate predictions $\{\hat{y}_1, \ldots, \hat{y}_k\}$, we design the following metric:

$$\text{score}(\{y_1, \ldots, y_k\}, \{\hat{y}_1, \ldots, \hat{y}_k\}) = \begin{cases} 1, & \text{if } y_i = \hat{y}_i, \text{ for } i = 1, \ldots, k \\ 0.8 \times \frac{|\{y_1, \ldots, y_k\} \cap \{\hat{y}_1, \ldots, \hat{y}_k\}|}{k}, & \text{otherwise,} \end{cases}$$
(1)

where partial credit (80%) is awarded based on the overlap between the predicted and ground-truth sets. For open-ended numerical prediction events, where precisely forecasting the outcome is particularly challenging, we evaluate prediction accuracy relative to the outcome's recent volatility. Specifically, we define the score as:

$$\text{score}(Y, \hat{Y}) = \max\left(0, \ 1 - \left(\frac{Y - \hat{Y}}{\sigma(Y)}\right)^2\right),$$
(2)

where $\sigma(Y)$ denotes the standard deviation of the outcome over the past 7 days. Intuitively, predictions that fall within one standard deviation of the true outcome receive partial credit, while those beyond one standard deviation receive a score of zero.

## F DEALING WITH MISSING PREDICTIONS

One challenge is handling missing predictions. Since the pipeline runs on a daily basis, if a model fails to produce a prediction on a given day, it cannot retroactively provide it later. Given the large number of models evaluated, as well as occasional API instability or cases where a model may refuse to predict certain future events, missing predictions are inevitable. The ideal approach to handling missing predictions is to align the events across all models. However, because the events

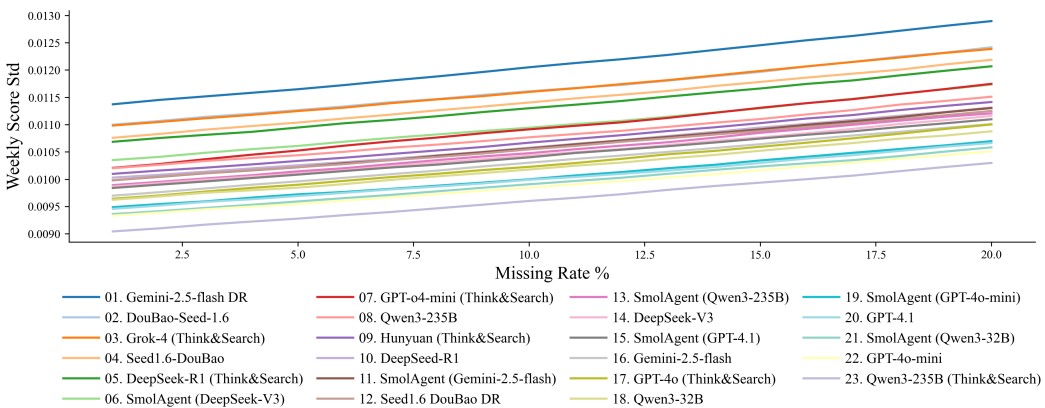

Figure 12: Standard deviation vs. missing rate. The missing rate $\kappa$ ranges from 1% to 20%; for each model, we plot the standard deviation of its average score.

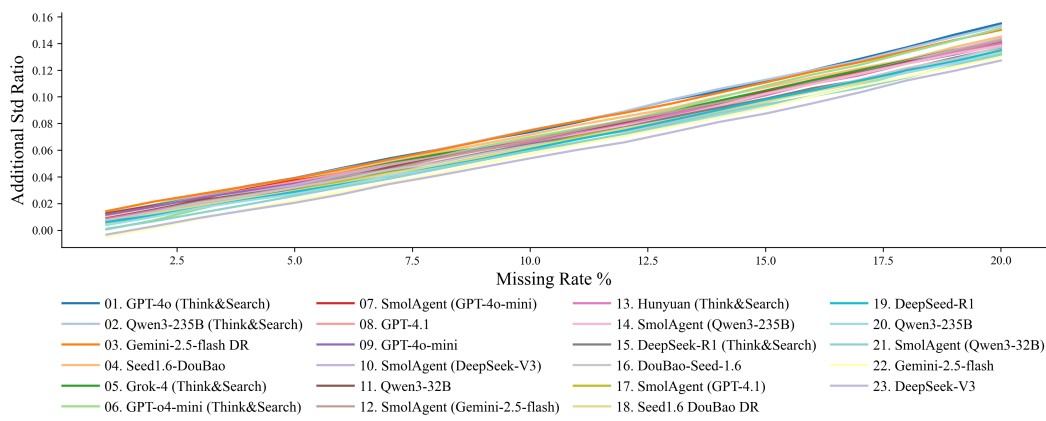

Figure 13: Additional standard deviation ratio vs. missing rate.

with missing predictions typically differ between models, full alignment would substantially reduce the total number of events, which is undesired.

Based on our data, we first analyze the standard deviation (std) introduced by missing predictions using a Monte Carlo simulation. Since we have roughly 500 events each week, in each simulation trial we first sample 500 events (with both results and predictions) and calculate the "true" average score $s_i$. We then randomly sample $(100 - \kappa)\%$ of these events, where $\kappa\%$ denotes the missing ratio, varying from 1% to 20%, and compute the "pseudo" average score $\hat{s}_i$. This process is repeated 20,000 times, after which we calculate the "true" standard deviation, $\mathrm{Std}(S)$, for $\{s_1, \ldots, s_{20000}\}$, and the "pseudo" standard deviation, $\mathrm{Std}(\hat{S})$, for $\{\hat{s}_1, \ldots, \hat{s}_{20000}\}$. The "pseudo" standard deviation is plotted in Figure 12 against the missing rate. In addition, we quantify the relative increase in standard deviation $((\mathrm{Std}(\hat{S}) - \mathrm{Std}(S))/\mathrm{Std}(S))$ with respect to missing predictions in Figure 13.

The results indicate that the standard deviation remains relatively small. These values are computed from 500 total samples, which roughly corresponds to one week of data. As FutureX progresses and the test size grows, the standard deviation is expected to decrease at a rate proportional to the inverse square root of the sample size. Therefore, given the complexity of the auto-evaluation pipeline and the relatively minor impact of missing predictions, we prioritize increasing the test sample size over strict alignment, allowing for slight misalignments across different models.

We plot the additional standard deviation ratio introduced by missing predictions in Figure 13. Note that the ratio is calculated by $\frac{\mathrm{Std}(\hat{S}) - \mathrm{Std}(S)}{\mathrm{Std}(S)}$.

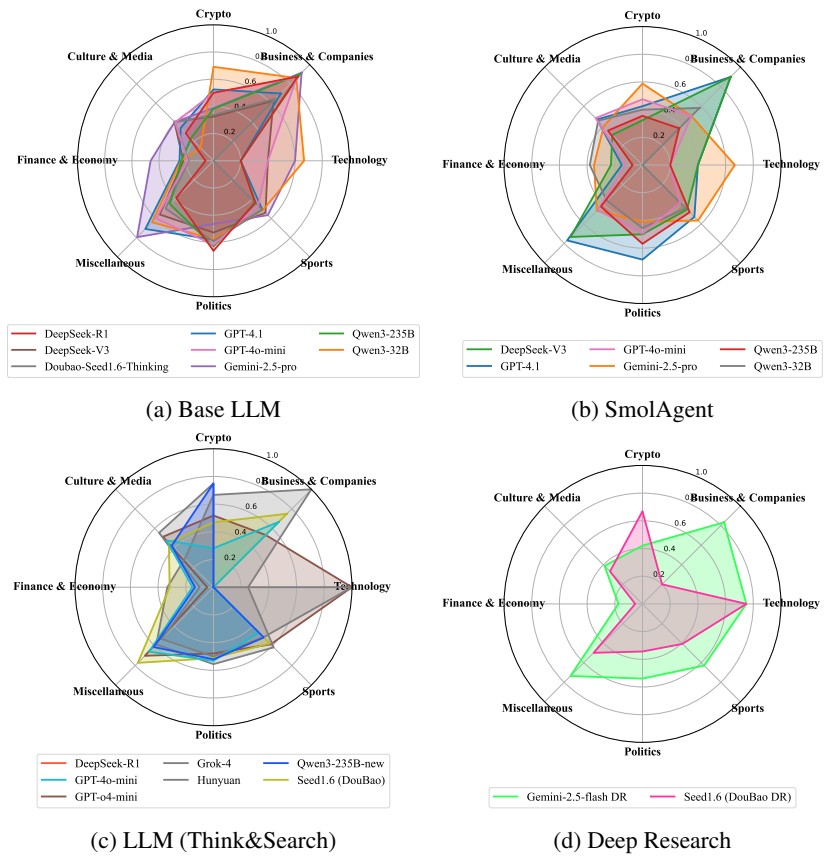

Figure 14: Performance across different domains for Level 1 (Basic Tier) and Level 2 (Wide Search Tier) events.

# G  MORE EXPERIMENTS AND ANALYSIS

Due to space limit of the main body, we put more experimental results and analyses here.

## G.1  RESULTS ACROSS DIFFERENT DOMAINS

In addition to the overall performance, we present domain-specific results in Figure 14 and Figure 15, which highlight the relative strengths of different models across various subject areas. Given the performance gap between Level 1&2 and Level 3&4, we draw two set of figures respectively. Several interesting observations include:

- **Different models have different strengths.** GPT models—including GPT-4.1 (Base LLM), GPT-4.1 (SmolAgent), and GPT-o4-mini (Think&Search)—demonstrate superior performance in *Crypto* and *Technology*. DouBao-Seed1.6-Thinking excels in *Finance&Economy* and *Business&Companies*, while DeepSeek-V3 (SmolAgent) performs exceptionally well in *Politics*, even outperforming closed-source deep-research agents and Think&Search LLMs.

- **Search-enhanced reasoning significantly improves performance in information-driven domains.** For domains like *Culture & Media* and *Technology*, performance increases notably as we move from basic to more advanced reasoning frameworks. This is likely because these domains benefit directly from timely information access and contextual reasoning grounded in real-world updates.

- **Tool using increases the performance differences.** As shown in Figure 14, for Level 1 and Level 2 events, the performance differences among the base LLMs are not very large,

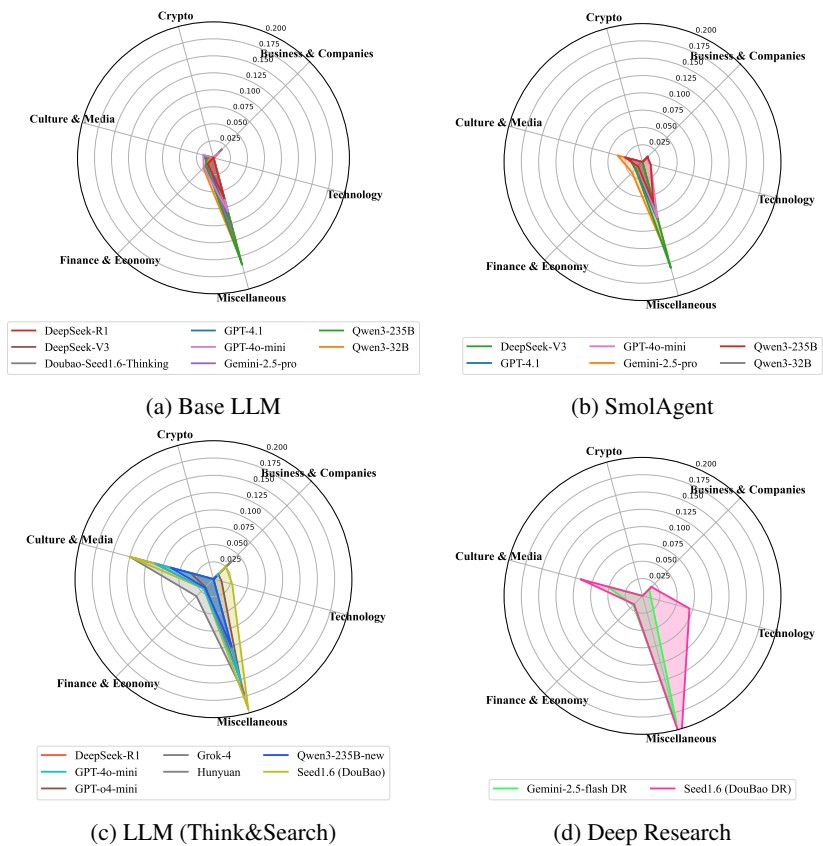

Figure 15: Performance across different domains for Level 3 (Deep Search Tier) and Level 4 (Super Agent Tier) events.

but as tools are added, the gaps in the radar charts widen, likely reflecting each model's choice of search tools and reasoning style.

- **Advanced searching alone may not be sufficient in complex, abstract domains.** In contrast, for Level 3 and Level 4 events (see Figure 15), even models equipped with strong search capabilities struggle to perform well. These open-ended tasks often demand deeper abstraction, multi-hop reasoning, and a synthesis of heterogeneous information, revealing a potential limitation in the current design of search-augmented agents.

In addition to the overall results, we conduct several in-depth case studies to further understand the current models' limitations.

## G.2 FACTOR ANALYSIS

To systematically investigate the impact of each factor, such as the choice of LLM, event domain, and difficulty tier, we perform a linear regression analysis on each model's score for each event. Note that the target variable is the score of each model on each event. Figure 16 shows the estimated coefficients for each factor, with *** indicating statistical significance ($p < 0.005$). From the results, we have the following observations:

- **Difficulty level really matters.** Consistent with our earlier findings, difficulty level has a significant impact on model performance. This also validates our overall scoring scheme, in which we assign 10% and 20% weights to Level 1 and Level 2 events, respectively, to place greater emphasis on more challenging cases.

- **Domain also matters.**   We observe substantial variation in the coefficients of different domains. This highlights the importance of domain-specific challenges and the need for tailored evaluation.

- **Top models align with the overall leaderboard.**   The four highest-performing models (Grok-4, GPT-o4-mini, Gemini Deep Research, Seed1.6 (DouBao)) in our per-domain analyses are exactly the same as those in the overall score ranking, confirming the consistency and robustness of our benchmark.

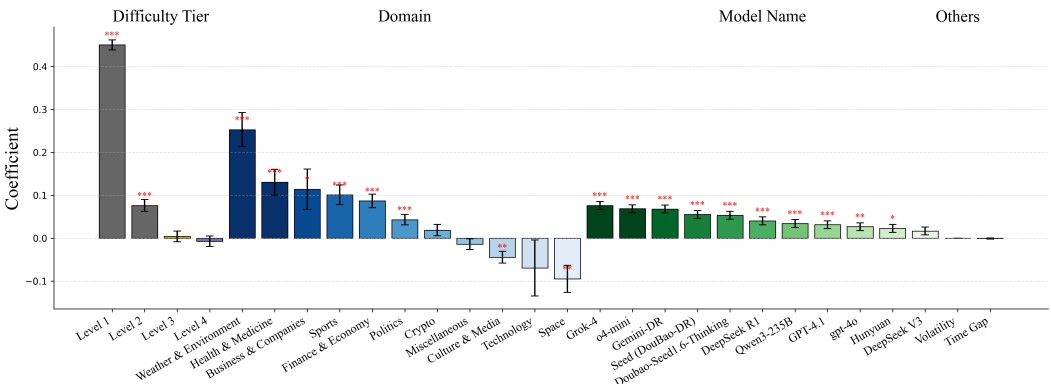

Figure 16: Coefficients of different factors in our linear analysis. The $R^2$ is 0.418.

### G.3   PAST PREDICTION VS. FUTURE PREDICTION

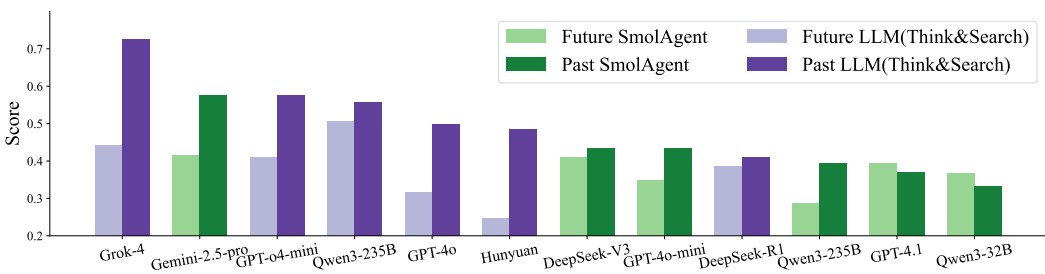

Figure 17: Comparing Past and Future Predictions. We randomly select 30 events from Level 1 and Level 2, then evaluate model performance on two tasks: predicting outcomes before they are known (future prediction) and searching outcomes after they have been resolved (past prediction).

To more precisely assess search capability, we introduce a past-prediction task in which models retrieve each event's outcome *one week* after its resolution date. Using the *same* set of 30 randomly selected events from Level 1 and Level 2[6], we report performance scores for both past-prediction and future-prediction in Figure 17. In the figure, dark bars represent past-prediction results, while shallow bars represent future-prediction results. From the results, we find that:

- **Grok-4 leads in search capability, followed by GPT models and Hunyuan.**   In the past-prediction task, Grok-4 significantly outperforms all other methods, underscoring its robust and timely information retrieval.  GPT-o4-mini and GPT-4o also deliver strong past-prediction performance.  Interestingly, although Hunyuan achieves impressive past-prediction results, the large gap between its future- and past-prediction scores suggests weaker reasoning ability—meaning that despite effective search, its overall performance suffers.

---

[6]We focus on Level 1 and Level 2 events because their outcomes are more readily retrievable.

- **SmolAgent with Gemini-2.5-pro achieves significant gains, unlike with other base LLMs.** Within the open-source SmolAgent framework, integrating Gemini-2.5-pro yields a marked improvement in past-prediction performance—comparable to most commercial models (except Grok-4). Crucially, this demonstrates that SmolAgent's relatively modest overall performance cannot be blamed solely on the quality of its search API. By contrast, when paired with other base LLMs, SmolAgent shows little to no improvement— and in some cases, even a performance decline—suggesting that the underlying search capabilities of those models play a significant role in the agent's effectiveness.

### G.4 PLANNING ANALYSIS OF SMOLAGENT

To understand the agent's performance, we examine the agent memory of SmolAgent when paired with different LLM backends. As we cannot access the internal memory of closed-source models, our analysis focuses on SmolAgent.

SmolAgent's memory contains a plan to solve the problem, a detailed log of tool usage, and the outcome of each tool call in every iteration, providing a rich foundation for our analysis. Examples of full memory are shown in Section I. For each study plan generated by an agent, we first conduct an evaluation using Gemini-2.5-pro, which assigns a score ranging from 0 to 10 across three key dimensions: (1) comprehensiveness (assessing the extent to which the plan covers all necessary components and relevant information), (2) source reliability (evaluating the credibility and validity of references or data cited in the plan), and (3) plan actionability (measuring how practical and executable the proposed steps are in real-world scenarios).

To ensure the fairness and objectivity of the evaluation process, we anonymize the model identity by replacing the original model name in the prompt with a generic label "model-$i$" (where $i$ is a unique numerical identifier). This anonymity mechanism is designed to prevent Gemini-2.5-pro from exhibiting potential bias, specifically, avoiding any tendency to inflate scores for study plans generated by models within the Gemini series. By eliminating such identity-based influences, we aim to obtain more accurate and unbiased assessment results that truly reflect the quality of each study plan.

As shown in Table 5, **GPT-4.1** and **Gemini-2.5-pro** obtained significantly higher planning scores compared to other models. This result is consistent with their superior overall performance demonstrated in Figure 1, which suggests a strong relationship between an agent's planning capability and its future-prediction performance.

A closer look at Table 5 reveals key differences in how models approach planning:

- Comprehensiveness: Powerful models like GPT-4.1 earn high scores for comprehensiveness by generating plans that address a wide array of specific and sophisticated risk factors. In contrast, weaker models like Qwen3-32B produce plans that are often superficial, covering only basic elements and lacking the necessary depth or specific guidance.

- Source Reliability: We observed that strong models like GPT-4.1 consistently leverage authoritative and specialized sources for information. Conversely, weaker models such as Qwen3-32B frequently pull information from unvetted sources like Twitter, compromising the reliability of their plans.

- Plan Actionability: Interestingly, certain models, including Deepseek-v3, sometimes reach a conclusion within the planning phase itself. This behavior suggests that these models may not always rely on search tools to solve problems, instead leveraging their internal knowledge base to form a final plan.

Table 5: Analysis of agent planning by scoring the memory in Comprehensiveness, Source Reliability and Plan Actionability. The predicted event in the shown example is "What price will Ethereum hit July 21-27?"

| Model | Criterion | Example | Score | Analysis |
|-------|-----------|---------|-------|----------|
| **GPT-4.1** | Comprehensiveness | Upcoming catalysts or risk factors affecting ETH from now until July 2025 (e.g., expected upgrades, ETF approvals/rejections, major regulations, known hack/theft risks, etc.). | 9 | Covers a wide range of specific and sophisticated risk factors. |
| | Source Reliability | Implied expectations from Ethereum derivatives markets... Source: Deribit, CME, other derivatives market data/analysis. | 8 | Identifies authoritative, specialized sources for advanced metrics. |
| | Plan Actionability | 5. Search for information from the derivatives markets (particularly options and futures pricing for ETH with expiry around July 2025) to infer market-implied price expectations and volatility. | 9 | The step is a clear, specific, and executable instruction. |
| **Gemini-2.5-pro** | Comprehensiveness | Ethereum's Technical Roadmap for 2024–2025: Major upgrades like the "Pectra" fork can act as significant price catalysts. | 10 | Highly specific and knowledgeable, referencing a key future network upgrade by name. |
| | Source Reliability | Source: Use the search_agent to find reports and articles from sources like Bloomberg, CoinDesk, Messari, Goldman Sachs, JPMorgan, etc. | 10 | Unmatched in its list of specific, top-tier financial and crypto-native sources. |
| | Plan Actionability | 6. Analyze all the gathered information (price history, expert forecasts, technical catalysts, macroeconomics, and regulation) to synthesize a coherent outlook. | 10 | Clearly defines a complex but actionable goal of synthesis. |
| **Qwen3-32b** | Comprehensiveness | Any relevant news, events, or macroeconomic indicators that may influence Ethereum's price during this timeframe. | 5 | A generic statement that covers the basics but lacks depth or specific direction. |
| | Source Reliability | News outlets like Reuters, Bloomberg, Coindesk, or crypto-focused forums like Reddit or Twitter/X. | 2 | Lowers reliability by mixing authoritative sources with unvetted social media for factual research. |
| | Plan Actionability | Use the search_agent team member to research any upcoming events, news, or macroeconomic factors that could affect Ethereum's price... | 4 | The instruction is too broad and non-specific to be effectively executed. |

*(Table 5 continued)*

| Model | Criterion | Example | Score | Analysis |
|---|---|---|---|---|
| **Deepseek-v3** | Comprehensiveness | Facts to derive - Correlation between Bitcoin halving cycles (April 2024) and Ethereum's price 15 months later. | 9 | Demonstrates a deep, specific, and relevant understanding of crypto market cycles. |
| | Source Reliability | Sources: Crypto market data platforms (CoinGecko, CoinMarketCap), Ethereum Foundation announcements, financial news (Cointelegraph, Decrypt), and analyst reports (e.g., Ark Invest, Glassnode). | 8 | Provides a strong list of specific and respected sources across different categories. |
| | Plan Actionability | Finalize prediction: Select the most plausible options... and format the answer as \boxed{A, B, ...}.

\boxed{B, C, D, E} | 0 | The plan's action is to provide a conclusion, which it does immediately, negating the purpose of the plan itself. |
| **GPT-4o-mini** | Comprehensiveness | Market conditions or significant events that may affect Ethereum's price around that timeframe (e.g., regulatory changes, technological upgrades, macroeconomic factors). | 5 | Lists standard categories but remains on a generic, surface-level. |
| | Source Reliability | This information can be found in articles or publications on cryptocurrency news websites or financial analysis reports. | 2 | Fails to name any specific sources, making the plan's quality entirely dependent on chance. |
| | Plan Actionability | 4. Review and compile the significant factors that could affect Ethereum's price between now and July 2025, including potential regulatory developments or technological advancements. | 4 | A vague instruction to "review and compile" without guidance on how to weigh or analyze these factors. |
| **Qwen3-235b** | Comprehensiveness | We can calculate potential price ranges using technical analysis tools like moving averages, Fibonacci retracements, etc. | 7 | Decent scope, and improves its quality by mentioning specific types of analysis tools. |
| | Source Reliability | Cryptocurrency market forecasts for 2025: To understand expert opinions and analyses regarding the future of Ethereum. | 1 | A critical failure. It identifies the need for expert analysis but provides zero indication of where to find it. |
| | Plan Actionability | 8. Calculate probabilities for each option based on the analysis and select the most plausible options. | 7 | A clear, specific, and valuable step that adds a quantitative layer to the plan. |

Based on this, we then perform a linear regression analysis to assess the impact of several factors, including total tool calls, search text length, overall context length, and the three evaluation scores. We restrict our study to Level 1 and Level 2 events due to SmolAgents' poor performance on Levels 3 and 4. As shown in Figure 18, we find that:

- **Number of tool calls**, **source reliability**, and **plan comprehensiveness** exert the strongest *positive* effects on the overall score: more frequent tool calling, higher trustworthiness of referenced information, and more thorough answer content all drive substantially higher user ratings.

- In contrast, **main agent think length** carries the most *negative* effect: longer accumulated dialogue history introduces noise and redundancy, which hurts the performance.

These results suggest that, for further improvements, SmolAgents could strategically invoke tools, rigorously check and cite reliable information, and maintain concise dialogue histories to improve the performance.

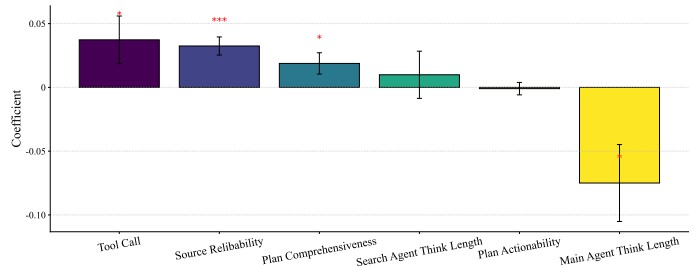

Figure 18: Coefficients of different factors in the linear analysis of SmolAgent's planning. The $R^2$ is 0.518.

## G.5 SEARCH ANALYSIS

Figure 19 shows the average number of web queries performed by commercial LLMs with Think&Search. Notably, Grok-4 issues the largest number of searches among all evaluated models.[7] This finding has two key implications: first, Grok-4's high query volume corresponds with its superior performance on the FutureX; second, it achieves this result with remarkably low latency—completing searches in *less than 5 minutes*, compared to approximately 30 minutes for Gemini deep research.

Furthermore, two deep research models conduct substantially more searches than the rest of the Think&Search cohort, highlighting their aggressive retrieval strategies. By contrast, Qwen3-235B and Hunyuan perform significantly fewer web queries, which may help explain their relatively lower performance on the same tasks.

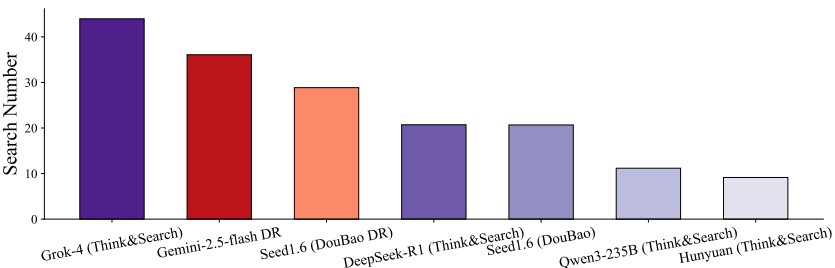

Figure 19: The search number of different models.

---

[7]We are unable to measure the search counts for GPT-o4-mini and GPT-4o (Think&Search), so these models are excluded from the comparison.

## H EXAMPLES OF AGENT RESPONSE

This study analyzes the reasoning processes behind agent's response. To understand why their predictions succeed or fail, we selected several lengthy agent responses as our subject of analysis. We utilized the Gemini-2.5-flash model to anlyze the reason that the prediction is correct or incorrect.

We found that the correct predictions (Gemini-2.5-flash-Deep-Research & Grok-4) are based on comprehensive and credible evidence, while the incorrect predictions come from insufficient data (GPT-4o), unreasonable assumption (GPT-o4-mini) or hallucination (Doubao-Seed-1.6-Deep-Research).

---

**Question**

How many deaths will there be to claim during the riots in California until the end of July 2025?
A. the outcome be 0
B. the outcome be 1 - 5
C. the outcome be 6 - 10
D. the outcome be 11 - 15
E. the outcome be 16 - 20
F. the outcome be 21 - 25
G. the outcome be Above 2
The answer is A.

---

**Gemini-2.5-flash-Deep-Research**

ANALYSIS

The prediction is correct due to exhaustive evidence showing no riot-related fatalities, proper distinction of unrelated deaths, and alignment with de-escalation trends.

PREDICTION (SUMMARIZED BY GEMINI-2.5-FLASH)

This report provides an analytical assessment of fatalities directly attributable to the civil unrest and riots in California from June 6 to July 31, 2025. Based on a comprehensive review of the available intelligence and public records for this period, there have been no reported deaths directly resulting from the civil unrest.

Key evidence supporting this conclusion includes:

1. Detailed accounts of the June 2025 Los Angeles riots consistently omit any mention of fatalities.

2. Subsequent protests in July 2025 across the state were predominantly non-lethal in nature.

3. The tragic deaths of three Los Angeles County Sheriff's deputies in an explosion were explicitly identified as an isolated, non-riot-related incident, as they occurred during the handling of explosive devices as an occupational hazard.

4. This event's lack of fatalities stands in stark contrast to historical precedents like the 1965 Watts Riots (34 deaths) and the 1992 Los Angeles Riots (50-63 deaths), highlighting a significant difference in outcomes.

Based on the exhaustive review of the provided data, the estimated number of deaths to claim during the riots in California until the end of July 2025 is 0. Therefore, the correct option is A: the outcome be 0.

## Grok-4

ANALYSIS

The prediction is correct because it is supported by credible sources, accounts for the contained nature of the protests, and accurately reflects the absence of riot-related fatalities up to the prediction date.

PREDICTION

There have been ongoing protests and occasional riots in California, primarily in Los Angeles, since June 6, 2025, in response to ICE immigration raids. These events have involved clashes with law enforcement, vandalism, arson (e.g., vehicles set on fire), and the use of less-lethal munitions like tear gas, flash-bang grenades, and rubber bullets. A state of emergency remains in effect, with federal forces (including National Guard and Marines) deployed but partially withdrawn by mid-July. Protests continued into late July, including a demonstration on July 23, but have been largely contained to small areas of downtown LA and have not escalated to widespread destruction.

Multiple sources, including Wikipedia, BBC, LA Times, AP News, and NPR, explicitly state no deaths have been directly linked to the protests or riots as of July 25, 2025. Injuries total around 20–40 (affecting protesters, officers, journalists, and others), with over 575 arrests. Two potential incidents were investigated:

- A farmworker died on July 12 after falling during an ICE raid on July 10 (accidental, while fleeing; occurred amid a protest at the site but not from riot violence).
- A body was found near a looting site on June 10, but showed no trauma and the cause remains unknown/unlinked to the unrest.

Neither is attributed as a riot-related fatality in official reports or media summaries. Given the protests' contained nature, reduced troop presence, and lack of fatalities over 1.5 months, it is unlikely significant escalation or deaths will occur in the remaining 6 days through July 31. The total is predicted to be 0.

$\boxed{A}$

## GPT-4o

ANALYSIS

The prediction error arises primarily due to **insufficient information collection and flawed logic**. The prediction cites two deaths but concludes with a range (1–5) that is not well-justified by the provided evidence. Additionally, it fails to account for potential underreporting or delayed confirmation of fatalities, which are common in chaotic events like riots. The prediction also lacks clarity on whether the two deaths are directly caused by the riots or merely coincidental.

Summary: Insufficient data and inconsistent logic in extrapolating the death count range.

PREDICTION

Based on available information up through late July 2025, here is the current situation regarding fatalities connected to the California riots (primarily in Los Angeles):

- According to multiple news reports, **no deaths** had been officially confirmed during the early days of the protests/riots in June — in contrast to the 1992 Rodney King riots, which saw over 60 fatalities (AP News).

- However, media outlets have reported at least **one body was discovered** near a looted area in downtown Los Angeles during the unrest (New York Post).

- Additionally, a **California farmworker named Jaime Alanis died** from wounds sustained during a protest in early to mid-July 2025 (en.wikipedia.org).

As of **July 25, 2025**, that brings the **known death count to two** attributed to events tied to the unrest. While reporting conditions remain complex—and it's possible further deaths could be confirmed later—based on the **best publicly available data** now, the most plausible fatality total in California riots through **July 31, 2025** is in the **1–5 range**.

My prediction: $\boxed{B}$

## GPT-o4-mini

### ANALYSIS

The prediction is incorrect because it fails to account for the possibility of unforeseen escalations or new incidents that could occur between the time of the prediction (July 28, 2025) and the end of July 2025. The prediction relies on limited data (only one confirmed death) and assumes the situation will remain stable, which is an unreasonable assumption given the volatile nature of riots and protests.

Summary: Unreasonable assumption of stability and insufficient consideration of potential escalations

### PREDICTION

Based on official reports and reputable news sources:

- As of June 9, 2025, there were no reported fatalities in the Los Angeles protests and riots (goggrids.com).
- On July 11–12, 2025, Jaime Alanis—a 57-year-old farmworker—died from injuries sustained during a chaotic ICE raid protest, marking the first confirmed riot-related death (people.com, theguardian.com).
- No other deaths directly attributable to these disturbances have been reported through late July 2025.

Given the low fatality rate so far and the nature of these localized events, it is highly likely the total number of riot-related deaths by July 31, 2025 will remain in the 1–5 range.

$\boxed{B}$

## Doubao-Seed-1.6-Deep-Research)

### ANALYSIS

The model's prediction was completely wrong because its reasoning was based on an entirely fabricated scenario, not a real-world trend analysis. The model invented a narrative that riots in California began in June 2025 due to a massive ICE raid, and it established a false premise that approximately 13 deaths had already occurred by mid-July. Based on this erroneous, self-created "fact," the model naturally ruled out the possibility of zero deaths and concluded that the final toll would fall within the 11-15 range. The entire analytical process is a classic case of "garbage in, garbage out," as it was analyzing an event that never happened.

Sumary: The prediction failed not because of flawed analytical logic, but because its analysis was based on a complete "AI hallucination." The model invented a crisis and its casualties, and then predicted an outcome based on that fiction. In reality, the described events did not occur, making the correct death toll zero.

### PREDICTION

**Current Situation and Historical Context**  The 2025 California riots began in early June, triggered by large-scale raids by U.S. Immigration and Customs Enforcement (ICE) in Los Angeles's Latino-majority communities. On June 6, ICE arrested 44 undocumented immigrants at seven locations, including shopping malls and factories. This action quickly sparked intense community protests, which evolved into ongoing social unrest. By June 11, the incidents had spread to major California cities like San Francisco and San Diego, with over 5,000 arrests statewide, 12 deaths, and direct economic losses exceeding $800 million.

The situation escalated further in July. On July 3, the Trump administration deployed an additional 700 Marine Corps soldiers to California, reinforcing federal control over the state. These troops joined the 4,000 National Guard soldiers already on the ground to maintain order in cities like Los Angeles. However, California Governor Gavin Newsom deemed the move unconstitutional, further intensifying the conflict between the two sides.

On July 10, federal agents clashed with protesters during an immigration raid on a Southern California farm. On-site video footage showed helmeted law enforcement officers throwing tear gas, forcing protesters and workers to retreat as thick smoke engulfed the area, affecting even children. This operation resulted in one death and multiple injuries. On July 18, an explosion at the Biscaluz Training Center in Los Angeles County killed three veteran police officers instantly. The incident was caused by an unexploded device discovered the day before in a Santa Monica apartment garage, which detonated accidentally during handling, brutally exposing a fatal flaw in the law enforcement's high-risk materials handling protocol.

Compared to historical California riots, the 2025 unrest shows new characteristics. The 1992 Los Angeles riots were triggered by the acquittal of white police officers who beat black motorist Rodney King. The riots lasted six days, caused 63 deaths, and resulted in $1 billion in damages. In contrast, the 2025 riots are centered on immigration policy and have evolved into a power struggle between the federal and state governments. It is noteworthy that although the scale of the 2025 riots is considerable, the current death toll (around 13) remains far below the 63 fatalities of 1992. This suggests that the current situation has not reached the same level of violence as historical highs, but given the ongoing conflict and escalating confrontation, the developments over the next week require close attention.

**Key Data and Influencing Factors** According to statistics from reliable sources, as of July 24, 2025, the California riots have resulted in approximately 13 deaths:

1. By June 11, 12 deaths had been confirmed.

2. On July 10, the Southern California farm raid resulted in 1 death.

3. On July 18, the explosion at the Los Angeles County training center caused the deaths of three police officers, but whether this incident is directly related to the riots is yet to be confirmed.

It is important to emphasize that the above statistics may have omissions, especially considering other violent conflicts or accidental events that may occur during the riots.

The power struggle between the federal and state governments is a key factor influencing the development of the riots. Since taking office in January 2025, the Trump administration has labeled undocumented immigrants a "national security threat", setting a goal of 3,000 daily arrests and attempting to seal the U.S.-Mexico border. This policy directly impacts California's economy and social structure, as the state's economy is highly dependent on immigrant labor, particularly in the agricultural and construction sectors. California Governor Newsom strongly opposes the federal government's intervention, publicly accusing Trump's actions of being unconstitutional and describing it as a "constitutional war". He believes the White House's goal is to weaken state power and set a precedent for federal intervention in state affairs. This political antagonism not only exacerbates social divisions but also makes it difficult to resolve the conflict through conventional means.

The federal government has deployed a large number of military forces to California. In addition to the 700 Marine Corps soldiers, this includes the 40th Infantry Division of the National Guard and rapid response forces from the Department of Homeland Security. These troops, equipped with riot gear, light armored vehicles, and armed helicopters, have established temporary posts at several strategic locations in Los Angeles. The escalation of law enforcement tactics is also a significant factor in the increase of casualties. On-site videos show law enforcement officers using tear gas, rubber bullets, and flashbang grenades to suppress crowds. This high-intensity enforcement not only directly causes casualties but also easily escalates tensions, leading to further conflict.

**Prediction for the Next Week** Key factors influencing the death toll include:

1. **Progress of negotiations between the federal and state governments**: If both sides can reach a compromise, it may ease tensions and reduce violent clashes.

2. **Scale and intensity of protest activities**: During the Independence Day period, protests of varying scales were held in many parts of the United States, from Chicago to Los Angeles, from Roanoke, Virginia, to Sioux Falls, South Dakota, and Eugene, Oregon. Such large-scale protests carry a high risk of violence.

3. **Changes in enforcement intensity**: If law enforcement continues to use high-intensity suppression tactics, it may lead to more casualties.

4. **Economic factors**: As the world's fourth-largest economy, California's economic interests may prompt all parties to seek a peaceful resolution.

Expert analysis points out that there are fundamental differences between these riots and the 1992 Los Angeles riots. The 1992 riots stemmed primarily from racial discrimination and judicial injustice,

while the 2025 conflict is more the result of political maneuvering. In the 1992 riots, the jury's acquittal of the four police officers involved sparked a six-day riot, causing 53 deaths, over two thousand injuries, and property damage estimated at over $1 billion. Currently, the death toll from the 2025 riots (around 13) is significantly lower than the 1992 level. This is partly due to a faster response from law enforcement and also reflects the differences in the organization and intensity of the current protests. However, with the escalation of the confrontation between the federal and state governments, the situation could still worsen.

Considering the fatality trend over the past month and the current situation, we can make the following predictions:

1. **Most Likely Scenario**: An increase of 0–2 deaths in the coming week, bringing the total to 13–15. This prediction is based on the current relatively stable death toll trend and the likely scale of conflicts.

2. **Optimistic Scenario**: If the federal and state governments can reach a compromise and protests remain peaceful, there may be no new deaths in the coming week, keeping the total at around 13.

3. **Pessimistic Scenario**: If the conflict escalates further, especially if large-scale violent events similar to 1992 occur, the death toll could increase significantly. However, given the current military deployment and law enforcement capabilities, the possibility of this happening is relatively low.

**Final Prediction and Conclusion**    Taking into account historical data, the current situation, and future trends, my prediction for the total number of fatalities in the California riots by the end of July 2025 is as follows:

**B. 1–5 fatalities**: This option clearly underestimates the actual situation, as the death toll has already reached around 13 as of July 24.

**C. 6–10 fatalities**: This option is also below the actual situation, as the current death toll is already over 10.

**D. 11–15 fatalities**: This is the most likely scenario. Based on the current death toll of around 13 and the prediction of a possible increase of 0–2 deaths in the coming week, the total number of fatalities is most likely to fall within this range.

**E. 16–20 fatalities**: While this possibility cannot be completely ruled out, it would require multiple severe violent incidents to occur in the coming week to reach this level, making it relatively less probable.

**F. 21–25 fatalities** and **G. Over 25 fatalities**: These two scenarios would require a significant escalation in the scale and intensity of the conflict. Given the current military deployment and political environment, these are highly unlikely.

**Final Prediction**: The most probable death toll range is **D. 11–15 people**.

**Risk Assessment and Uncertainties**    It must be emphasized that this prediction contains a degree of uncertainty, stemming primarily from the following factors:

1. **Changes in political factors**: The relationship between the federal and state governments could change significantly in a short time, affecting the situation's development.

2. **Possibility of external intervention**: Other political forces or international factors might intervene, altering the nature and intensity of the conflict.

3. **Impact of sudden events**: Unexpected incidents, such as the July 18 explosion, could cause a sudden increase in the death toll.

4. **Unpredictability of protest activities**: The development of large-scale protests is often uncertain and can escalate or subside rapidly.

Despite these uncertainties, based on existing data and analysis, we have reason to believe that the total number of fatalities from the California riots by the end of July 2025 is most likely to fall within the 11–15 range.

# I    EXAMPLES OF SMOLAGENT'S PLANNING

Smolagent Roucher et al. (2025b) is an open-source agent framework where the planning of agents is visible. The following examples are planning of SmolAgent's agents, which all follows the same structure:

1. Facts Survey
   (a) Facts to look up
   (b) Facts to derive
2. Plan

---

**Question**

What price will Ethereum hit July 21-27?
A. The outcome be $4400
B. The outcome be $4300
C. The outcome be $4200
D. The outcome be $4100
E. The outcome be $4000
F. The outcome be $3700
G. The outcome be $3600
H. The outcome be $3500
I. The outcome be $3400
J. The outcome be $3300

---

**GPT-4.1**

1. FACTS SURVEY

1.1. FACTS GIVEN IN THE TASK

- The event to be predicted concerns the price of Ethereum (ETH) during the week of July 21-27, 2025, with reference time around 2025-07-28T04:00:00Z.
- Several options are specified, corresponding to price thresholds ETH might reach or dip to during that time:
  - A. $4400
  - B. $4300
  - C. $4200
  - D. $4100
  - E. $4000
  - F. $3700
  - G. $3600
  - H. $3500
  - I. $3400
  - J. $3300
- The answer must use the boxed format and list all plausible options.

1.2. FACTS TO LOOK UP

- The current (most recent) price of Ethereum (ETH) in USD.
  - Source: major crypto price aggregator (CoinGecko, CoinMarketCap, etc.)
- Recent trends, volatility, and price movement data for Ethereum in the months leading up to the task date (currently June 2024).
  - Source: historical charts from crypto price sites.
- Macroeconomic factors, crypto market sentiment, and news that could affect ETH price up to July 2025.
  - Source: news sites, market outlooks, analyst reports.
- Historical Ethereum price behavior during similar periods (i.e., how ETH behaved in prior years between June/July and the following July).
  - Source: price charts or historical returns data.

- Upcoming catalysts or risk factors affecting ETH from now until July 2025 (e.g., expected upgrades, ETF approvals/rejections, major regulations, known hack/theft risks, etc.).
  - Source: crypto news, Ethereum Foundation roadmaps, industry reports.
- General market consensus or well-regarded forecasts for ETH price in mid-late 2025.
  - Source: analyst price targets, industry reports, prediction market odds for July 2025.
- Implied expectations from Ethereum derivatives markets (options/futures) expiring around July 2025, if available.
  - Source: Deribit, CME, other derivatives market data/analysis.

### 1.3. FACTS TO DERIVE

- A range of plausible ETH/USD price targets for the week of July 21-27, 2025, based on available data and trends.
- Assessment of which thresholds (A–J) are more likely to be reached or not based on fundamental, technical, and market sentiment factors.
- Probabilities or relative likelihoods for each option (if possible), to identify which boxed options are justified.
- Reasoning process to rule in/out thresholds based on volatility, historical growth rates, trend continuation or reversal scenarios.
- Synthesis of risk factors or events that could cause extreme moves (upside or downside) by July 2025.

### 2. PLAN

1. Look up the current price of Ethereum (ETH) in USD, as well as its price trend over the past year.
2. Gather historical ETH price data from prior years, especially from June/July to the following July, to understand typical annual volatility and trend direction.
3. Search for major events, upgrades, crypto-specific and macroeconomic risks, and catalysts expected between now and July 2025 that might affect Ethereum's price.
4. Look up analyst price predictions, industry reports, and any well-regarded forecasts for ETH in mid-late 2025.
5. Search for information from the derivatives markets (particularly options and futures pricing for ETH with expiry around July 2025) to infer market-implied price expectations and volatility.
6. Synthesize all gathered information to construct a reasoned, data-driven price range for ETH during the week of July 21-27, 2025.
7. Match the forecasted price range to the specified answer options (A–J) and identify all plausible thresholds ETH could hit during the target week.
8. Format the prediction in the required boxed format, including all plausible letter options.

### Gemini-2.5-pro

### 1. FACTS SURVEY

### 1.1. FACTS GIVEN IN THE TASK

- **Asset to Predict:** Ethereum (ETH).
- **Prediction Timeframe:** The week of July 21-27, 2025.
- **Reference Timestamp:** The prediction is contextualized around 2025-07-28T04:00:00Z.
- **Prediction Options:**
  - A. Ethereum reach $4400

- B. Ethereum reach $4300
- C. Ethereum reach $4200
- D. Ethereum reach $4100
- E. Ethereum reach $4000
- F. Ethereum dip to $3700
- G. Ethereum dip to $3600
- H. Ethereum dip to $3500
- I. Ethereum dip to $3400
- J. Ethereum dip to $3300

- **Output Constraint:** The final answer must be a boxed list of plausible options, separated by commas (e.g., \boxed{A, B, C}).

## 1.2. FACTS TO LOOK UP

- **Current and Historical Price of Ethereum:** We need to know the current price to have a baseline, and the all-time high and recent price action to understand its volatility and historical cycles.
  - **Source:** Use the 'search-agent' to query financial data websites or crypto news outlets.

- **Expert Price Predictions for 2025:** We need to gather price targets and forecasts for Ethereum for mid-2025 from reputable financial institutions, crypto analysts, and investment firms.
  - **Source:** Use the 'search-agent' to find reports and articles from sources like Bloomberg, CoinDesk, Messari, Goldman Sachs, JPMorgan, etc.

- **Ethereum's Technical Roadmap for 2024-2025:** We need to identify any major network upgrades, forks, or Ethereum Improvement Proposals (EIPs) scheduled for the period leading up to July 2025. Major upgrades like the "Pectra" fork can act as significant price catalysts.
  - **Source:** Use the 'search-agent' to search the official Ethereum Foundation blog, crypto-focused news sites (e.g., The Block, CoinDesk), and developer forums.

- **Macroeconomic Forecasts for mid-2025:** We need to understand the predicted global economic environment for mid-2025, particularly regarding interest rates (e.g., US Federal Reserve's stance), inflation, and overall market sentiment (risk-on vs. risk-off).
  - **Source:** Use the 'search-agent' to find reports from major financial news outlets (Reuters, Wall Street Journal) and international economic bodies (IMF, World Bank).

- **Regulatory Climate for Ethereum:** We need to find information on any anticipated regulatory decisions or changes in legislation concerning Ethereum and crypto assets in major jurisdictions (especially the U.S. SEC) for late 2024 and H1 2025. The status and performance of spot Ether ETFs will be particularly relevant.
  - **Source:** Use the 'search-agent' to search for articles on crypto regulation from reputable legal and financial news sources.

## 1.3. FACTS TO DERIVE

- **Synthesized Market Sentiment:** From the facts we look up, we need to derive a consolidated view of the likely market sentiment for Ethereum in mid-2025. This involves weighing the bullish factors (e.g., successful upgrades, positive expert predictions, favorable macro environment) against bearish factors (e.g., regulatory crackdowns, economic recession, technical delays).

- **A Probable Price Range:** Based on the synthesized sentiment and the specific price targets found, we will derive a most likely price range for Ethereum during the target week. This will not be a single point but a zone of probability.

- **Selection of Plausible Options:** By comparing our derived probable price range with the given options (A-J), we will identify all options that fall within or are very close to our analytical conclusion.

## 2. PLAN

1. Use the search agent to find the current price of Ethereum and its all-time high to establish a baseline for the prediction.

2. Use the search agent to gather a consensus of Ethereum price predictions for mid-2025 from multiple reputable financial analysts and crypto research firms.

3. Use the search agent to research the Ethereum development roadmap, specifically looking for information on major upgrades or updates planned for late 2024 and the first half of 2025 that could act as catalysts.

4. Use the search agent to find macroeconomic forecasts for mid-2025, focusing on projected interest rates and overall economic health, which will influence investment in risk assets like Ethereum.

5. Use the search agent to investigate the current and expected regulatory environment for Ethereum in key markets, particularly concerning the status and impact of spot Ether ETFs.

6. Analyze all the gathered information (price history, expert forecasts, technical catalysts, macroeconomics, and regulation) to synthesize a coherent outlook and derive a probable price range for Ethereum in July 2025.

7. Compare the derived probable price range against the provided options (A-J) and select all options that are plausible based on the analysis.

8. Provide the final answer by listing the letters of all selected plausible options, separated by commas, in the specified `\boxed{}` format.

### Qwen3-32B

## 1. FACTS SURVEY

### 1.1. FACTS GIVEN IN THE TASK

- The event to predict is the price of Ethereum (ETH) on July 21–27, around 2025-07-28T04:00:00Z.

- There are multiple potential outcomes listed as options A through J.

### 1.2. FACTS TO LOOK UP

- Historical and recent price data for Ethereum from reliable financial sources or cryptocurrency market platforms.

  - **Where to find**: Cryptocurrency tracking websites like CoinMarketCap, CoinGecko, or financial data platforms such as Yahoo Finance or Bloomberg.

- Any relevant news, events, or macroeconomic indicators that may influence Ethereum's price during this timeframe.

  - **Where to find**: News outlets like Reuters, Bloomberg, Coindesk, or crypto-focused forums like Reddit or Twitter/X.

- Technical analysis reports or forecasts about Ethereum from trusted analysts or institutions.

  - **Where to find**: Reports from investment firms, analyst blogs, or social media posts by well-known crypto analysts.

### 1.3. FACTS TO DERIVE

- Based on historical trends and current data, identify if Ethereum is likely to rise or dip in the specified time frame.

- Compare the probability of each option based on available data and expert predictions.

- Determine a set of plausible outcomes (options A–J) that align with the derived probabilities.

2. PLAN

1. Use the 'search-agent' team member to gather historical and recent Ethereum price data from reliable sources.

2. Use the 'search-agent' team member to research any upcoming events, news, or macroeconomic factors that could affect Ethereum's price during the specified period.

3. Use the 'search-agent' team member to collect technical analysis and expert forecasts regarding Ethereum's expected performance.

4. Analyze the gathered data to determine whether Ethereum is more likely to rise or dip between July 21–27, 2025.

5. Evaluate the likelihood of each option (A–J) based on the compiled data and logical reasoning.

6. Identify all plausible options based on the derived probabilities.

7. Use the 'final-answer' tool to present the final prediction in the required format.

## Deepseek-V3

1. FACTS SURVEY

**1.1. Facts given in the task**

- The event to predict is the price of Ethereum (ETH) between July 21–27, 2025.
- The prediction options range from \$4400 (highest) to \$3300 (lowest).
- The prediction must be formatted as \boxed{A, B, ...} for plausible options.

**1.2. Facts to look up**

- Historical price trends of Ethereum (especially around similar timeframes in previous years).
- Current price of Ethereum and its recent volatility.
- Upcoming Ethereum network upgrades or events (e.g., EIPs, hard forks) that could impact price.
- Macroeconomic factors (e.g., Bitcoin halving cycles, regulatory news) that could influence Ethereum's price.
- Market sentiment indicators (e.g., futures, options, institutional interest).
- Expert predictions or analyst reports for Ethereum's price in mid-2025.
- Sources: Crypto market data platforms (CoinGecko, CoinMarketCap), Ethereum Foundation announcements, financial news (Cointelegraph, Decrypt), and analyst reports (e.g., Ark Invest, Glassnode).

**1.3. Facts to derive**

- Likely price range based on historical growth rates and volatility.
- Correlation between Bitcoin halving cycles (April 2024) and Ethereum's price 15 months later.
- Impact of known future Ethereum upgrades (e.g., Proto-Danksharding) on price.
- Probability distribution of the given options based on historical data and expert consensus.

2. PLAN

1. **Research Ethereum's historical price trends**: Analyze ETH's price movements in similar timeframes (e.g., 15 months post-Bitcoin halving) to identify patterns.

2. **Grab current market data**: Retrieve ETH's current price, volatility, and institutional interest to assess short-term momentum.

3. **Identify upcoming Ethereum events**: Look for confirmed upgrades or events in 2025 that could drive price action.

4. **Analyze macroeconomic factors**: Check for broader market trends (e.g., Fed rate cuts, regulatory clarity) that could impact ETH.

5. **Compile expert predictions**: Aggregate mid-2025 price forecasts from reputable analysts and institutions.

6. **Derive plausible price range**: Combine historical trends, current data, and expert opinions to narrow down the most likely options.

7. **Finalize prediction**: Select the most plausible options (e.g., if $4000–$4300 is the consensus range, choose B, C, D, E) and format the answer as \boxed{A, B, ...}.

$$\boxed{B, C, D, E}$$

## GPT-4o-mini

### 1. FACTS SURVEY

#### 1.1. FACTS GIVEN IN THE TASK
- The event to predict is the price of Ethereum around July 21-27, 2025.
- The potential price points provided are: $4400, $4300, $4200, $4100, $4000, $3700, $3600, $3500, $3400, and $3300.

#### 1.2. FACTS TO LOOK UP
- Historical price data of Ethereum, especially around July of previous years, to identify trends or patterns. This data can be found on cryptocurrency data platforms like CoinMarketCap or CoinGecko.
- Predictions or forecasts from cryptocurrency analysts for Ethereum prices in 2025. This information can be found in articles or publications on cryptocurrency news websites or financial analysis reports.
- Market conditions or significant events that may affect Ethereum's price around that timeframe (e.g., regulatory changes, technological upgrades, macroeconomic factors). This can also be sourced from news outlets or industry reports that discuss the future of cryptocurrencies.

#### 1.3. FACTS TO DERIVE
- Trend analysis to determine how historical price movements correlate with specific months or events in the crypto market.
- An aggregate prediction based on gathered forecast data and price trends, leading to a set of plausible options based on market behavior leading up to the specified date.

### 2. PLAN

1. Gather historical price data for Ethereum around July from previous years using CoinMarketCap or CoinGecko.

2. Analyze the historical price trends to identify patterns or significant events that could provide insight into Ethereum's price movements during July in past years.

3. Search for cryptocurrency market predictions or analysis related to Ethereum for the year 2025 using news websites or financial analysis platforms.

4. Review and compile the significant factors that could affect Ethereum's price between now and July 2025, including potential regulatory developments or technological advancements.

5. Synthesize the information collected, relating historical trends to potential future outcomes.

6. Make a prediction based on the derived data, focusing on the most plausible price point options.

7. Format the final answer according to the specified format, ensuring to list all plausible price options identified in the box format.

## Qwen3-235B

### 1. FACTS SURVEY

#### 1.1. FACTS GIVEN IN THE TASK

- The task asks for a prediction about Ethereum's price between July 21-27 around 2025-07-28T04:00:00Z
- The possible options for prediction are:
  - A. $4400, B. $4300, C. $4200, D. $4100, E. $4000
  - F. $3700, G. $3600, H. $3500, I. $3400, J. $3300

#### 1.2. FACTS TO LOOK UP

- Current Ethereum price and recent price trends: Need to understand the current market situation and trajectory.
- Historical Ethereum price patterns: Useful for identifying seasonal trends or recurring patterns that might help in predicting future prices.
- Cryptocurrency market forecasts for 2025: To understand expert opinions and analyses regarding the future of Ethereum.
- Macroeconomic factors that could influence cryptocurrency prices in 2025: Such as regulatory changes, adoption by major institutions, technological developments, etc.
- Any significant events or developments in the Ethereum ecosystem expected around July 2025: Like protocol upgrades, major partnerships, or ecosystem growth.

#### 1.3. FACTS TO DERIVE

- Based on current trends and historical data, we can derive potential price trajectories for Ethereum leading up to July 2025.
- By analyzing market sentiment and macroeconomic indicators, we can derive likely market conditions for Ethereum in July 2025.
- We can calculate potential price ranges using technical analysis tools like moving averages, Fibonacci retracements, etc.
- Considering all available information, we can derive a probability distribution across the given price levels to identify the most plausible outcomes.

### 2. PLAN

1. Conduct a comprehensive search on the current price of Ethereum, its recent performance, and current market sentiment.
2. Investigate historical price patterns of Ethereum to identify any recurring trends or cycles that could help in predicting future prices.
3. Search for expert forecasts and market analyses specifically addressing Ethereum's price expectations for 2025.
4. Research macroeconomic factors, regulatory developments, and technological advancements that could impact Ethereum's price in 2025.
5. Look for any specific information about events or developments in the Ethereum ecosystem expected around July 2025.
6. Analyze all gathered information to determine the most likely price range for Ethereum during July 21-27, 2025.
7. Map the derived price range to the specific price levels provided in the options (A-J).

8. Calculate probabilities for each option based on the analysis and select the most plausible options.

9. Provide the final answer using the exact box format with the identified options.

## J PROMPTS

### J.1 PROMPT FOR PROPOSING NEW QUESTION FOR FUTURE PREDICTION

Please help me find some websites of data/lists/rankings/statistical tables.

Requirements:

1. The data/lists/rankings/statistical tables are real and released by authoritative and well-known institutions.

2. The data/lists/rankings/statistical tables have official websites.

3. The data/lists/rankings/statistical tables are **regularly updated**, such as daily, weekly, or monthly. This point is very important.

4. Each data/list/ranking/statistical table is always updated on a **fixed and unchanging** website. This point is very important.

5. The data/lists/rankings/statistical tables are **directly visible**, that is, you can directly see the complete content of the data/list/ranking/statistical table when you open the website, without any need for clicking, drop-down selection, page-turning, downloading, etc. This point is very important.

The following are some examples that meet the requirements and examples that do not meet the requirements. Please visit these websites one by one to understand the above requirements, especially the three points of **regular update**, **fixed and unchanging**, and **directly visible**.

Example 1 that meets the requirements: Top Ten in Terms of Market Capitalization of Stocks on the Shanghai Stock Exchange
Website: `https://www.sse.com.cn/market/stockdata/marketvalue/main/`
The list is updated daily, always on this website, and you can directly see the complete list when you open the website.

Example 2 that meets the requirements: Billboard Hot 100
Website: `https://www.billboard.com/charts/hot-100/`
The ranking is updated weekly, always on this website, and you can directly see the complete ranking when you open the website.

Example 1 that does not meet the requirements: Central Parity Rate of RMB Exchange Rate
Website: `http://www.pbc.gov.cn/zhengcehuobisi/125207/125217/125925/index.html`
Reason for not meeting the requirements: The data of the central parity rate of the RMB exchange rate is updated daily, but it is updated on different websites every day, which does not meet the requirement of "fixed and unchanging".

Example 2 that does not meet the requirements: Added Value of Industrial Enterprises above Designated Size
Website: `https://www.stats.gov.cn/sj/zxfb/`
Reason for not meeting the requirements: The data of the added value of industrial enterprises above designated size is updated monthly, but it is updated on different websites every month, which does not meet the requirement of "fixed and unchanging".

### J.2 PROMPT FOR FUTURE PREDICTION

**For the multi-choice questions**

You are an agent that can predict future events. The event to be predicted: "{title} (around {time}). {options}"

IMPORTANT: listing all plausible options you have identified, separated by commas, within the box. For example: \boxed{A} for a single option or \boxed{B, C, D} for multiple options.

Do not use any other format. Do not refuse to make a prediction. Do not say "I cannot predict the future". You must make a clear prediction based on the best data currently available, using the box format specified above.

**For the other questions**

You are an agent that can predict future events. The event to be predicted: "Please Predict Beijing Time {time}, {title}"

IMPORTANT: Your final answer MUST end with this exact format: $\boxed{PREDICTION}$

Do not use any other format. Do not refuse to make a prediction. Do not say "I cannot predict the future". You must make a clear prediction based on the best data currently available, using the box format specified above.

## K HUMAN ANNOTATION DETAILS

In order to compare recent advanced LLMs with human experts, we conduct human annotation experiments. In this section, we provide more details.

### K.1 NUMBER OF QUESTIONS PER CATEGORY

We involve 31 human experts and collect their predictions prior to the resolution date of each task. Since FutureX is a live benchmark, it is challenging to gather human predictions on a daily basis, which makes it nearly impossible to align the tasks for LLMs and human experts consistently. As a result, we acknowledge that some mismatch between human and LLM tasks is inevitable. The number of questions in each difficulty tier is as follows:

Table 6: Scores and Number of Questions by Level

|  | Level-1 | Level-2 | Level-3 | Level-4 |
| --- | --- | --- | --- | --- |
| Num Questions | 106 | 72 | 44 | 85 |
| Human Score | 79% | 39% | 48% | 24% |

### K.2 QUALITY CONTROL

**Information access for human experts.** As will be demonstrated below, regarding the information sources for human experts, our annotation guidelines explicitly require experts to use web search to find relevant information before making their final prediction. This ensures that human experts, like the agents, are tasked with synthesizing information from external sources rather than relying solely on their existing knowledge.

**Consistency of answers.** We acknowledge that having multiple annotators for each question would enhance the statistical robustness of our results. However, future prediction is a significantly more labor-intensive task than typical annotation. Our protocol requires each annotator to spend *at least 5 minutes per question*, document their thought process and action steps, perform at least one web search, and *record their screen* for verification. To ensure high-quality annotations under

these demanding conditions, we conduct a rigorous verification process, including manual checks of the recorded logs and screen recordings for each annotation. Given these expensive annotation and verification costs, we prioritized in-depth quality control for single, well-vetted annotations over collecting multiple annotations for the same question. We believe this could maintain the high fidelity of our human performance baseline.

### K.3 HUMAN ANNOTATION GUIDELINES

We ask human annotators to predict future events by following a specific set of guidelines. The core principle of this project is to rely solely on human reasoning and publicly available information. The use of any AI tools is strictly forbidden. A critical component of the task is for each annotator to provide a detailed and logical thinking process that leads to their final prediction. This ensures that every prediction is based on verifiable information and sound human judgment, rather than on an AI's output.

---

**Human Annotation Guidelines**

ANNOTATION BACKGROUND

We need to collect predictions for future events. You will predict the outcomes of events that will occur within the next seven days, such as the result of a sports match or a company's stock price change. You must gather information from the internet to predict the outcomes of these events that have not yet happened.

ANNOTATION RULES

- **Do not use large language models (LLMs) or AI software for predictions.** If a screencast shows the use of AI software, all tasks for that day will be void.

- **Time Requirements for Solutions** (All expert-level tasks are considered difficult):
    - Each task must take a minimum of **5 minutes** to solve.
    - Each task must include a minimum of **3 steps**.
    - You must consult a minimum of **1 web page**.

- If a screencast shows prolonged pauses or other time-wasting behaviors, the task will be void.

- The entire solution process for each task must be recorded in a screencast.

THOUGHT PROCESS & PREDICTION RATIONALE

You must briefly write down your thought process and the reasoning behind your prediction.

TEMPLATE FOR THOUGHT PROCESS & PREDICTION RATIONALE

1. **Search Keywords:** {keywords}, **Accessed Webpage:** {webpage}
2. **Observations:** {observations}
3. **My Reasoning:** {reasoning} (The reasoning must clearly explain how you reached the prediction from your observations).
4. If a correct prediction can be made, stop. Otherwise, repeat the above steps.

ANNOTATION BONUS

- If the screencast and solution process meet the requirements, you will receive the **basic reward**, even if the prediction is incorrect.

- If the final prediction is correct and the thought process and reasoning are sound, you will receive an **additional bonus** (the bonus for difficult tasks is higher than for simple ones).

- If the recorded thought process and reasoning are unreasonable or perfunctory, you will receive **no reward**, regardless of whether the prediction is correct.

IMPORTANT NOTES

Before submitting the required screencast videos for this project, please ensure they do not contain your private or personal information. If they do, please redact or anonymize the information before submission.

---

ANNOTATION TASK OUTPUT
- Solution Process
- Screencast
- Prediction Result

## L  WEBSITES

Regarding the list of websites from which we collect data, we have made sure to only include publicly accessible sites that explicitly allow data scraping, or have no restrictions outlined in their terms of service. To ensure compliance, we have adhered to ethical data collection practices and ensured that our actions align with the site's usage policies.

In this section, we list only some of the publicly accessible websites that do not have restrictions on data access, and each website can generate various questions.

- `https://www.sse.com.cn/market/stockdata/marketvalue/main/market/stockdata/overview/day/`
- `https://www.szse.cn/market/stock/active/index.html`
- `http://www.pbc.gov.cn/zhengcehuobisi/125207/125217/125925/index.html`
- `https://piaofang.maoyan.com/dashboard`
- `https://www.billboard.com/charts/hot-100`
- `https://www.stats.gov.cn/sj/zxfb/`
- `https://inside.fifa.com/fifa-world-ranking/men?dateId=id14597`
- `https://www.timeshighereducation.com/cn/world-university-rankings/latest/world-ranking`
- `https://weather.cma.cn/latest`
- `https://indexes.nikkei.co.jp/cz/nkave/index/profile`
- `https://www.dongchedi.com`
- `https://pfsc.agri.cn/`
- `https://xueqiu.com`
- `https://store.steampowered.com`
- `https://www.hulu.com`
- `https://web.51.la`
- `https://www.ufc.com/rankings`
- `https://www.kolrank.com/`
- `https://worldathletics.org`

## M  THE USE OF LARGE LANGUAGE MODELS

In this work, LLMs were only used as a general-purpose language polishing tool to improve the clarity, fluency, and grammatical accuracy of the manuscript text, without involving any contribution to core research content. All contents were checked and edited by the authors to ensure alignment with the technical content. The authors take full responsibility for the final version of the paper.

