# OpenReview forum: "FutureX: An Advanced Live Benchmark for LLM Agents in Future Prediction"
_ICLR.cc/2026/Conference — ICLR 2026 Poster_

### Official Review · Reviewer_S6CR · 2025-10-31

**Soundness:** 4
**Presentation:** 4
**Contribution:** 2
**Rating:** 8
**Confidence:** 3

**Summary:**

The paper introduces FutureX, a large-scale live benchmark designed to evaluate LLM agents on future-prediction tasks that require reasoning, information gathering, and decision-making. Unlike prior static benchmarks, FutureX uses a semi-automated pipeline that continuously crawls, curates, and resolves questions from 195 websites across 11 domains. The system automatically collects future-oriented questions, gathers LLM predictions at event start dates, and scores them after resolution, enabling a contamination-free, real-time evaluation. The benchmark assesses 25 models, from base LLMs to research agents (e.g., Grok-4, Gemini Deep Research), with human-expert comparisons.

**Strengths:**

- Contamination-free design: The design of the closed environment to prevent information contamination is well-executed and addresses a critical challenge in LLM evaluation.
- Comprehensive model evaluation: The study includes 25 models spanning reasoning, search, and deep-research agents with human expert baseline, providing a comprehensive evaluation landscape.

**Weaknesses:**

- The pipeline's automated expansion might favor scale over fidelity. No quantitative validation is provided to ensure the "extra" questions truly add value.
-  The paper lists 195 websites but will be good to include the full list of the selected websitesl. More details on domain balance, filtering reliability would help.
- The authors may consider including Brier score as most prior works in forecasting adopt this metric, which would facilitate comparison with existing literature.

**Questions:**

- What are the daily compute and maintenance costs for running FutureX, given the live crawling and 25-model evaluation pipeline?
- Could future versions compare LLM predictions with crowd forecasts (e.g. prediction-market aggregates) to test alignment with collective prediction?
- Will the full list of websites, question templates, and answer-extraction code be released for transparency and reproducibility?

---

> ### Author Response · Authors · 2025-11-19
>
> We thank Reviewer S6CR for the positive feedback and the excellent suggestions for improving the transparency and rigor of our work.
>
> ### **Q1: The pipeline's automated expansion might favor scale over fidelity.**
>
> We appreciate this concern. To ensure fidelity, our automated pipeline includes several quality control steps, which are detailed in **Appendix C** of our submission. These include:
> *   For **Level-1 and Level-2 questions**, we use betting odds and an LLM-as-Judge to filter out low-quality or trivial questions.
> *   For **Level-3 and Level-4 questions**, we manually curate the source websites to exclude those unsuitable for prediction.
>
> ### **Q2: Include the full list of the selected websites and more details.**
>
> Thank you for the suggestion. Details on our event filtering process and domain balance are available in **Appendix C and D** of our submission. And we have added the publicly accessible websites that do not have restrictions on data access in **Appendix L**.
>
>
> ### **Q3: Consider including Brier score.**
>
> Thank you for this constructive suggestion. We agree that the Brier score is a valuable metric. Our initial focus was on establishing a robust benchmark for predictive accuracy, following the practice of many recent agent benchmarks. We view the incorporation of probabilistic metrics like the Brier score as a crucial next step for FutureX .
>
> ### **Q4: What are the daily compute and maintenance costs for running FutureX?**
>
> Thank you for this very practical and important question. Given our automated pipeline for question and answer crawling, the operational cost of FutureX is primarily driven by API calls, which is directly proportional to the evaluation frequency.
> The cost consists of two parts, the initial experiments presented in this paper and the cost of long-term maintenance.
>
> 1. **For the Experiments in This Paper:** To generate a substantial number of data points for a robust analysis across all 25 agents, we conducted our evaluations on a **daily prediction cycle**. This intensive schedule, while providing rich results for our study, is indeed computationally expensive, requiring tens of thousands of API calls per day.
> 2. **For Long-Term Maintenance:** We recognize that a daily evaluation cycle would be prohibitively expensive for most research groups, including us own, to sustain over the long term. Therefore, to ensure the longevity and accessibility of FutureX as a public resource, after the release, we run the evaluations on a **weekly basis**. This reduces the computational cost to a manageable level (tens of thousands of API calls per week), making it an affordable commitment for us.
>
> ### **Q5: Could future versions compare LLM predictions with crowd forecasts (e.g. prediction-market aggregates) to test alignment with collective prediction?**
>
> That is an excellent suggestion, and we thank you for pointing out this fascinating direction for future research. We completely agree that comparing agent predictions to crowd forecasts is a valuable evaluation dimension.
>
> First, we would like to clarify our current approach to human evaluation, which shares a similar motivation. We recognized that a strong human baseline was essential. However, since our **Level-3 and Level-4 questions are generated from news sources and do not exist on prediction markets**, we could not use market aggregates for the entire benchmark. To ensure a consistent human baseline across all four difficulty levels, we recruited **human experts** to provide predictions for every question. This provided a robust, albeit labor-intensive, human performance benchmark.
>
> This brings us to your brilliant point about using prediction-market aggregates. For our **Level-1 and Level-2 questions**, which are sourced directly from these markets. We are very excited about this idea and plan to incorporate this comparison for Level-1 and Level-2 questions in the next iteration of FutureX.  Thank you again for this valuable suggestion.
>
>
> ### **Q6: Will the full list of websites, question templates, and answer-extraction code be released for transparency and reproducibility?**
> We add some of the publicly accessible websites that do not have restrictions on data access in **Appendix L**, and we've uploaded the evaluation code. As for the question templates, we will release the past questions after accepted.

---

### Official Review · Reviewer_Kgce · 2025-10-31

**Soundness:** 3
**Presentation:** 3
**Contribution:** 3
**Rating:** 6
**Confidence:** 4

**Summary:**

This paper introduces FutureX, a large-scale, dynamic benchmark for evaluating the future prediction capabilities of LLM agents. The authors argue that existing benchmarks fail to test complex reasoning and decision-making under real-world uncertainty.

FutureX uses a fully automated pipeline that continuously gathers future-oriented questions from 195 diverse websites. It runs 25 different LLM agent models to get their predictions on a "start date." After the event's "resolution date," the system automatically finds the ground-truth outcome and scores the predictions. This live-updating design inherently eliminates data contamination, as the answers do not exist at the time of prediction.

The benchmark includes four difficulty tiers, ranging from simple single-choice questions (Level 1) to high-volatility, open-ended numerical and ranking tasks (Level 4). The paper's findings show that while agents with search tools (like Grok-4) perform best, all models struggle significantly with the harder tiers. Furthermore, all evaluated agents still perform substantially worse than a baseline of 40 human experts.

**Strengths:**

The primary strength of FutureX is its novel "live" evaluation design. By focusing on future events whose outcomes are not yet known, it provides a robust and scalable solution to the critical problem of data contamination that plagues static benchmarks.

The automated pipeline for question collection and answer verification is a significant technical achievement, ensuring the benchmark remains current and challenging.

Another strength is the logical stratification of tasks into four difficulty tiers. The results validate this structure, showing a clear performance drop as tasks move from simple retrieval (Level 1/2) to complex, open-ended reasoning under uncertainty (Level 3/4).

I like the the inclusion of a human expert baseline. It provides crucial context, grounding the model scores and highlighting the significant gap that still exists between current AI agents and human-level analytical reasoning.

**Weaknesses:**

As far as I understand, the benchmark's "prediction window." is only "one-week". This restricts the evaluation to short-term predictions, not long-term forecasting. It fails to test an agent's ability to reason about events months or years in the future, which is a different and critical skill for human analysts. I think this heavily limited this benchmark.

The evaluation metrics, while appropriate for correctness, do not capture the probabilistic nature of forecasting. The benchmark does not assess an agent's ability to express calibrated confidence (e.g., providing a probability or a confidence interval). It only scores the accuracy of a single-point answer unlike some recent works that assess the confidence too.

The comparison to human experts is a bit ambiguous. It is not specified whether the 40 human experts and the 25 LLM agents had access to the exact same information retrieval tools (I think this is very important as the author pointed out about potential leakage of data to models; such leakage could happen to human experts too), making the performance gap difficult to interpret fairly.

the benchmark tests the entire agent system at once. This makes it difficult to isolate the point of failure. When an agent performs poorly, it's unclear whether the fault lies with the underlying LLM's reasoning or with the agent's planning and tool-use framework.

**Questions:**

What are the computational and financial costs associated with maintaining this live benchmark? Its complexity might make it difficult for other researchers to reproduce or build upon.



How does the benchmark disentangle the performance of the base LLM from the performance of the agent framework? For example, are the failures of open-source agents due to poor planning logic or the base model's inability to follow the plan?

Are there plans to expand the benchmark's scope beyond the one-week prediction window to include long-term forecasting tasks, which require different reasoning skills than short-term information synthesis?

---

> ### Author Response · Authors · 2025-11-19
>
> Thank you for the positive feedback and thoughtful questions. We address your concerns as follows:
>
> ### **Q1: The benchmark's prediction window is only "one-week".**
>
> This is an excellent point. We chose a one-week prediction window as a deliberate starting point for several strategic reasons:
>
> 1.  **Practical Value:** Weekly predictions hold significant value and are common in real-world applications, such as on prediction market websites.
> 2.  **Sufficient Challenge:** Our results show that this one-week window is **already a significant challenge** for current state-of-the-art agents.
> 3.  **Actionable Feedback Loop:** The rapid development cycle of LLMs demands timely feedback. A weekly cycle provides actionable insights that align with the pace of model iteration, whereas a monthly or yearly cycle would introduce significant lag.
>
> Importantly, our evaluation framework is **flexible and not inherently tied to a one-week window**. It can be readily extended to longer time horizons, and we do plan to host questions with longer time horizons in the future.
>
> ### **Q2: The benchmark does not assess an agent's ability to express calibrated confidence.**
>
> Thank you for this valuable suggestion. We agree that incorporating calibrated confidence scores would create a more comprehensive evaluation. Our decision to use accuracy as the primary metric was guided by the standards in many contemporary agent benchmarks (e.g., BrowserGym, SWE-bench), which focus on task correctness. We consider the integration of confidence scores an excellent direction for future work and plan to incorporate it in the next iteration of FutureX.
>
> ### **Q3: Ambiguity in the comparison to human experts' information access.**
>
> Thank you for raising this important point. As detailed in **Appendix K.3 (page 41, Human Annotation Guidelines)**, the human annotation guidelines instructed experts to use internet search. We did not restrict them to a specific search engine. We acknowledge that ensuring identical tools for all parties is practically challenging, especially since the backends of proprietary agents are undisclosed. However, we can confirm that the open-source frameworks we used (**SmolAgents and AgentOrchestra**) both utilize **Google Search**. Our setup ensures both humans and agents perform a similar task: actively seeking and synthesizing information from the open web.
>
> Thanks again for your suggestions, and we have revised our **Appendix K** to provide more details of human annotation.
>
> ### **Q4: Difficulty in isolating the point of failure.**
>
> This is an excellent question that gets to the heart of agent evaluation. We agree that a key function of a good benchmark is to help disentangle the sources of failure.
> Due to space limits, we put more analyses as well as diagnostic implications in **Appendix G**, which provide a framework for diagnosing whether poor performance stems from the underlying LLM's reasoning, the agent's planning, or its tool-use framework.
>
> 1. **Diagnosing Search and Tool-Use:** To isolate search capability, we conducted a "retrospective search" experiment (detailed in **Appendix G.3**). We tasked agents with answering questions about events that had already occurred. By comparing this performance to their predictive performance, we could assess their tool-use effectiveness. We found that top agents like Grok-4 were effective at retrospective search, suggesting their predictive failures are more related to reasoning under uncertainty rather than a fundamental inability to find information. In contrast, weaker agents failed at both tasks, pointing to core issues in their tool-use framework.
> 2. **Diagnosing Planning and Reasoning:** To diagnose planning abilities, we used an LLM-based evaluation to score the planning phase of agent trajectories on three key metrics: **Comprehensiveness, Source Reliability, and Plan Actionability** (detailed in **Appendix G.4**). As shown in **Table 5**, there is a clear gap between strong and weak models across these metrics, demonstrating that FutureX can effectively pinpoint deficiencies in an agent's planning logic.
> 3. **Quantifying Performance Factors:** To provide a more quantitative diagnosis, we analyzed the correlation between various behavioral factors and final prediction performance (**Figure 18, Appendix G.4**). Our findings indicate that the **number of tool calls, source reliability, and plan comprehensiveness** are the most significant contributors to an agent's success.
>
> Taken together, these analyses provide a framework for diagnosing agent weaknesses.

---

> > ### Author Response · Authors · 2025-11-19
> >
> > ### **Q5: Computational and financial costs associated with maintaining this live benchmark?**
> >
> > Thank you for this very practical and important question. Given our automated pipeline for question and answer crawling, the operational cost of FutureX is primarily driven by API calls, which is directly proportional to the evaluation frequency.
> > The cost consists of two parts, the initial experiments presented in this paper and the cost of long-term maintenance.
> >
> > 1. **For the Experiments in This Paper:** To generate a substantial number of data points for a robust analysis across all 25 agents, we conducted our evaluations on a **daily prediction cycle**. This intensive schedule, while providing rich results for our study, is indeed computationally expensive, requiring tens of thousands of API calls per day.
> > 2. **For Long-Term Maintenance:** We recognize that a daily evaluation cycle would be prohibitively expensive for most research groups, including us own, to sustain over the long term. Therefore, to ensure the longevity and accessibility of FutureX as a public resource, after the release, we run the evaluations on a **weekly basis**. This reduces the computational cost to a manageable level (tens of thousands of API calls per week), making it an affordable commitment for us.
> >
> > ### **Q6: How does the benchmark disentangle the performance of the base LLM from the performance of the agent framework?**
> >
> > As discussed in our response to **Q4**, our analyses in the appendix are designed to shed light on this. Our analysis of performance factors confirms that both tool-use and planning quality are crucial, allowing for a more nuanced diagnosis of failure points.
> >
> > ### **Q7: Are there plans to expand the benchmark's scope beyond the one-week prediction window?**
> >
> > Yes, absolutely. Our framework is designed to be flexible. We started with a one-week window because it is already very challenging for current agents. We plan to extend the benchmark to include **monthly or even longer-term forecasting tasks**, which will require significantly extended durations. This is a straightforward extension of our existing pipeline.

---

### Official Review · Reviewer_diMG · 2025-11-02

**Soundness:** 3
**Presentation:** 3
**Contribution:** 3
**Rating:** 8
**Confidence:** 3

**Summary:**

This paper presents FutureX, a large-scale, live benchmark for evaluating base LLMs and LLM agents on real-world future prediction tasks. It features a four-stage pipeline for event database construction, future event curation, agent prediction, and answer acquisition, with all stages executed daily to ensure dynamic, contamination-free evaluation across diverse domains and difficulty levels. Experiments with 25 models show that search- and tool-augmented agents outperform base LLMs but still lag behind humans on complex tasks, establishing FutureX as a scalable framework for assessing LLM reasoning in uncertain, real-world environments.

**Strengths:**

1. Future prediction is a good testbed for evaluating the capabilities of LLMs and agents in information gathering, reasoning, and predictive analysis, while naturally mitigating data contamination since ground-truth answers are not available yet at prediction time.

2. FutureX provides an automated and scalable pipeline for data construction, future event curation, and answer verification, offering good practical value with minimal manual effort.

3. The tiered event categorization in FutureX (Basic, Wide Search, Deep Search, Super Agent) enables systematic assessment of reasoning depth and tool usage in LLM agents.

4. The evaluation spans 25 models covering base, search-augmented, and tool-augmented agents, which offers comprehensive and comparative insights across different model classes.

5. The paper is clearly written and well-organized, with informative figures and tables.

**Weaknesses:**

1. Some details on human annotation are unclear. For example, the number of questions per category, whether human experts had access to all information sources or relied solely on their own knowledge, and the consistency of their answers on the same question. Such information would clarify the human–agent performance comparison and also serve as an indicator of the robustness and quality of the questions in the benchmark.

2. While the performance analysis across 25 models is comprehensive, the paper offers limited discussion on the diagnostic utility of FutureX. It is not clear whether the benchmark can help identify specific weaknesses in agents or provide actionable insights for improving their reasoning and prediction capabilities. Such discussion will further benefit the benchmark users and the development of new agents.

**Questions:**

1. In Line 252, it is mentioned that "the answer acquisition success rate exceeds 97%". How is this success rate calculated?

---

> ### Author Response · Authors · 2025-11-19
> **Response to Reviewer diMG**
>
> We sincerely thank Reviewer diMG for the positive feedback and insightful questions, which have helped us improve the quality of our manuscript.
>
> **Q1: Some details on human annotation are unclear.**
>
> Thank you for your suggestions. We provide more details of human annotation as follows, which has been added to our revision version (see **Appendix K**).
>
> 1.  **Number of questions per category:** We have now added a detailed breakdown of the number of questions for each difficulty level in a new table in the Appendix. We have referenced this table in the main body of the paper.
>
> |               | Level-1 | Level-2 | Level-3 | Level-4 |
> | ------------- | ------- | ------- | ------- | ------- |
> | Num Questions | 106     | 72      | 44      | 85      |
> | Score         | 79%     | 39%    | 48%    | 24%    |
>
>
> 2.    **Information access for human experts:** Regarding the information sources for human experts, our annotation guidelines, detailed in **Appendix K.3 (page 41, Human Annotation Guidelines)**, explicitly required experts to use web search to find relevant information before making their final prediction. This ensures that human experts, like the agents, were tasked with synthesizing information from external sources rather than relying solely on their existing knowledge.
> 3.   **Consistency of answers:** We agree that having multiple annotators for each question would enhance the statistical robustness of our results. However, future prediction is a significantly more labor-intensive task than typical annotation. Our protocol requires each annotator to spend **at least 5 minutes per question**, document their thought process and action steps, perform at least one web search, and **record their screen** for verification. To ensure high-quality annotations under these demanding conditions, we conduct a rigorous verification process, including manual checks of the recorded logs and screen recordings for each annotation. Given these expensive annotation and verification costs, we prioritized in-depth quality control for single, well-vetted annotations over collecting multiple annotations for the same question. We believe this could maintain the high fidelity of our human performance baseline.
>
> **Q2: Limited discussion on the diagnostic utility of FutureX.**
>
> Thank you for the insightful comment. We agree that demonstrating FutureX's ability to identify specific agent weaknesses is a crucial contribution. Due to space limits, we put more analyses in **Appendix G**, and we will move this to the main body for the final version (with the extra page).
> In general, our analysis reveals a strong correlation between agent behaviors and their prediction performance. Specifically:
>
> 1. **Diagnosing Search Capability:** In **Appendix G.3**, we isolate the agent's search ability. We tasked agents with answering questions about events that had already occurred within the past week (a "retrospective search" task). By comparing this with their performance on predicting the same events a week prior, we found that top-performing agents like **Grok-4** is also the best in past-events prediction, with their retrospective accuracy being more than double their predictive accuracy. This suggests their limitation is not in finding relevant information per se, but perhaps in synthesizing incomplete information for prediction as well as the reasoning based on the existing information. Conversely, weaker agents (e.g., those based on SmolAgent) performed poorly on both tasks, indicating fundamental issues with their tool-use and search strategies.
> 2. **Diagnosing Planning and Reasoning:** In **Appendix G.4**, we further analyze the agents' planning and reasoning abilities. We used an LLM-based evaluation to score the planning phase of agent trajectories on three key metrics: **Comprehensiveness, Source Reliability, and Plan Actionability**. As shown in **Table 5**, there is a clear and consistent gap between strong and weak models across these metrics, demonstrating that FutureX can effectively pinpoint deficiencies in an agent's planning logic.
> 3. **Quantifying Performance Factors:** To provide a more quantitative diagnosis, we analyzed the correlation between various behavioral factors and final prediction performance (**Figure 18, Appendix G.4**). Our findings indicate that the **number of tool calls, source reliability, and plan comprehensiveness** are the most significant contributors to an agent's success.
>
> Taken together, these analyses demonstrate that FutureX can offer actionable insights by diagnosing specific weaknesses in an agent's search, planning, and reasoning capabilities, thereby guiding future development efforts.

---

> > ### Author Response · Authors · 2025-11-19
> >
> > **Q3: How is the "answer acquisition success rate" calculated?**
> >
> > Thank you for the question. The "answer acquisition success rate" refers to the percentage of questions in the FutureX benchmark for which our automated system successfully retrieved a ground-truth answer after the event occurred. This process involves dynamic web crawling and information extraction. Since websites can change their structure or content, this process can occasionally fail. Through careful website filtering and prompt optimization, we have stabilized this success rate at over 97%, ensuring a reliable and continuous evaluation process.

---

### Meta-Review · Area_Chair_jPtD · 2026-01-08

**Summary:**

FutureX presents a valuable contribution to LLM agent evaluation by introducing a large-scale, live benchmark for future prediction tasks. The paper addresses the critical problem of data contamination through an automated pipeline that collects questions before answers are available, evaluates 25 models, and provides comprehensive results. Reviewers unanimouslyappreciated the novel "live" design, clear presentaton, comprehensive model evaluation, and practical vaue. Key concerns raised include: (1) limited discusson on diagnostic utility initially (authors added sustantial analysis in appendix), (2) one-week prediction window being restrictive (authors note flexibility and plan extensions), (3) unclear human annotatio details (clarified in revision), (4) lack of probabilistic metrics like confidence calibration and Brier scores (acknowledged as future work), and (5) difficulty isolating failure points (addressed with diagnostic analyses). The authors provided thorough rebuttals with detailed appendix additions addressing most concerns. Overall, the work makes a solid contribution to agent evaluation benchmarking.

**Reviewer Concerns:**

Addressed: (1) Limited diagnostic utility - authors added substantial analyses in Appendix G demonstrating FutureX can isolate search capability, planning/reasoning abilities, and quantify perfrmance factors; (2) Unclear human annotation details -revision clarifies question distribution per category, human expert search protocols, and verification procedures; (3) Answer acquisition success rate definition - clarified as 97%+ success in automated answer retrieval; (4) Scale over fidelity concern - authors documented quality control steps (betting odds filtering, LLM-as-Judge for Levels 1-2, manual curation for Levels 3-4).

Outstanding: (1) One-week prediction window limitation - authors acknowledge this focuses on short-term forecasting; expansion to longer horizons is noted as planned but not yet implemented. (2) Probabilistic metrics - benchmark lacks confidence calibration and Brier scores; authors identify this as future work. (3) Computational costs - addressed with discussion of weekly maintenance at manageable cost, though daily experimental costs noted as expensive. (4) Full website list and reproducibility - some proprietary webste details withheld, though publicly accessible list provided in ppendix and evaluation code released.

**Reviewer Scores:**

Reviewer diMG (Xinyu Li): Initial rating 8 (accept oster) would likely remain at 8 or increase given coprehensive rebuttals addressing annotation clarity and diagnostic capability oncerns.
Reviewer Kgce (Zhangdie Yuan): Initial ratig 6 (marginally above acceptance) would likely increse to 7-8 based on authors' responses clarifying diagostic utility in appendix, human annotation procedurs, and framework flexibility for future extensions. However, fundamental concerns about one-wek window and lack of confidence metrics may limit upside.
Reviewer S6CR (Elvis Hsieh): nitial rating 8 (accept poster) would likely remain t 8 or potentially increase given thorough responseson quality control mechanisms, website transparency,future incorporation of market aggregates, and code elease plans.
Overall, two reviewers rate 8 (clear acept), one rates 6-7 (lean accept with concerns). Authors' detailed rebuttals strengthen cases but do ot fully resolve longer-term forecasting scope and probabilistic metric limitations.

---

### Decision · Program_Chairs · 2026-01-26

Accept (Poster)